# Identification of the Notch ligand DLK1 as an immunotherapeutic target and regulator of tumor cell plasticity and chemoresistance in adrenocortical carcinoma

Nai-Yun Sun [1], Suresh Kumar[1], Yoo Sun Kim[1], Diana Varghese[1], Arnulfo Mendoza[2], Rosa Nguyen [2], Reona Okada [2], Karlyne Reilly [2], Brigitte Widemann [2], Yves Pommier [1], Fathi Elloumi[1], Anjali Dhall[1], Daiki Taniyama[1], Mayank Patel[3], Etan Aber[2], Cristina F. Contreras[2], Rosandra N. Kaplan[2], Katja Kiseljak-Vassiliades [4,5], Margaret E. Wierman[4], Dan Martinez[6], Jennifer Pogoriler [6], Amber K. Hamilton[7], Sharon J. Diskin[7], John M. Maris [7], Robert W. Robey [8], Michael M. Gottesman[8], Jaydira Del Rivero[1] & Nitin Roper[1] ✉

While immunotherapeutic targeting of cell surface proteins is an increasingly effective cancer therapy, identification of new surface proteins, particularly those with biological importance, is critical. Here, we uncover delta-like non-canonical Notch ligand 1 (DLK1) as a cell surface protein with limited normal tissue expression and high expression in multiple refractory adult metastatic cancers including small cell lung cancer (SCLC) and adrenocortical carcinoma (ACC), a rare cancer with few effective therapies. In ACC, ADCT-701, a DLK1 targeting antibody-drug conjugate (ADC), shows in vitro and in vivo activity but is overall limited due to high expression and activity of the drug efflux protein ABCB1 (MDR1, P-glycoprotein). In contrast, ADCT-701 induces complete responses in DLK1+ ACC and SCLC in vivo models with low or no ABCB1 expression. Genetic deletion of DLK1 in ACC dramatically downregulates ABCB1 and increases ADC payload and chemotherapy sensitivity through NOTCH1-mediated transdifferentiation. This work identifies DLK1 as an immunotherapeutic target that regulates tumor cell plasticity and chemoresistance in ACC and supports an active phase I clinical trial targeting DLK1 with an ADC in ACC and neuroendocrine neoplasms (NCT06041516).

Targeting cell-surface antigens with antibody-drug conjugates (ADCs) is a promising immunotherapeutic approach in oncology with recent FDA approvals across a diverse set of malignancies[1]. Nonetheless, identification of new tumor-specific targets is imperative, especially for refractory adult metastatic tumors with few treatment options, and for less common malignancies, such as neuroendocrine (NE) neoplasms, with unique biological features.

One defining feature of NE neoplasms, as the categorization of these cancers implies, is NE differentiation, characterized by high expression of a coordinated set of genes, including synaptophysin and

chromogranin, routinely used as clinical diagnostic markers for these tumors. The Notch pathway is a major negative regulator of NE differentiation, and suppression of this pathway is common across NE tumors[2]. While mechanisms of Notch pathway suppression in NE cancers are not entirely clear, it is known that cell surface Notch ligands such as delta-like ligand 3 (DLL3) inhibit Notch pathway activation in normal development[3]. Moreover, as DLL3 expression is restricted to the brain but aberrantly expressed in many NE cancers, DLL3 was an early immunotherapeutic target in small cell lung cancer (SCLC)[4] and NE prostate cancer[5]. While initial efforts to target DLL3 with an ADC failed, more recent efforts targeting DLL3 via T-cell engager strategies in SCLC have demonstrated remarkable success[6,7] with recent approval by the FDA for treatment of relapsed SCLC.

In this work, we sought to assess whether Notch ligands beyond DLL3 may or may not be targetable cell surface proteins in cancer. We screened normal tissue and metastatic cancer datasets for expression of Notch ligands (*DLL1*, *DLL3*, *DLL4*, *DLK1*, *JAG1*, *JAG2*) and uncovered DLK1 (delta-like non-canonical Notch ligand 1) as a candidate cell surface immunotherapy target protein. Moreover, we show that DLK1 is targetable by an ADC, particularly in the rare cancer adrenocortical carcinoma (ACC) in which DLK1 is highly expressed. Importantly, we find that DLK1 is a key driver of chemoresistance in ACC through regulation of NOTCH1 signaling and the drug efflux protein ABCB1 (MDR1, P-glycoprotein), thereby demonstrating an important biological function for this new immunotherapeutic target.

## Results

### DLK1 has limited normal tissue expression and high expression in multiple metastatic cancers including adrenocortical carcinoma

To assess whether Notch ligands could be suitable cell surface immunotherapeutic targets, we compared normal tissue expression of *DLL1*, *DLL4*, *DLK1*, *JAG1*, and *JAG2* with *DLL3* using the adult Genotype-Tissue Expression (GTEx) Portal[8]. As expected, expression of *DLL3* was restricted to the brain (Supplementary Fig. 1A). However, other Notch ligands (*DLL1*, *DLL4*, *JAG1*, and *JAG2*) were expressed across a wide span of normal tissues (Supplementary Fig. 1A) except *DLK1*, which was expressed in only the adrenal gland, pituitary, ovary, hypothalamus, and testis (Supplementary Fig. 1B). We next assessed for tumor expression of *DLK1* using RNA-seq data from a cohort of ~1000 adult patients with treatment refractory metastatic cancers[9]. We observed high *DLK1* expression in a subset of these refractory cancers, such as sarcomas, SCLC, germ cell tumors, and grade 2 NE tumors (Fig. 1A). High *DLK1* expression has also been recently observed in pediatric neuroblastoma[10]. Strikingly, almost all treatment refractory metastatic adrenal cancers, i.e., ACC and pheochromocytoma/paraganglioma (PCPG), expressed high levels of *DLK1* (Fig. 1A), which we also observed among primary adrenal tumors in the TCGA PanCancer dataset[11] (Fig. 1B). While both ACC and PCPG are rare tumors of the adrenal gland (ACC incidence of ~ 0.5-2 cases per million people per year and PCPG incidence of 2-8 cases per million people per year[12]), we focused further analysis on ACC as it is an aggressive, highly malignant cancer with an overall poor prognosis (5-year survival 38%[13]) with an urgent need for new treatment options[14]. *DLK1* was the most highly expressed Notch ligand with little to no expression of *DLL3* across multiple ACC cohorts[15,16] (Fig. 1C and Supplementary Data 1 and 2). To validate these RNA-seq data, we performed DLK1 IHC across our cohort of ACC metastatic tumors and found 97% (*n* = 28/29) of ACC patients were DLK1⁺ by cytoplasmic/membranous scoring (mean H-score 147, range 10-300) reflecting a mix of intensity and distribution of DLK1 in ACC (Fig. 1D), which is similar to recent DLK1 IHC data from a large cohort of ACC tumors[17]. Thus, our data demonstrate DLK1 as a potential new surface immunotherapeutic target in multiple malignancies, particularly ACC.

### ADCT-701, an antibody drug conjugate targeting DLK1, induces cytotoxicity in ACC through apoptosis and bystander killing

Given the high and near ubiquitous expression of DLK1 in ACC, we next sought to determine if DLK1 could be targeted in ACC using a DLK1-directed antibody-drug conjugate (ADCT-701)[10,18]. ADCT-701 consists of HuBA-1-3D, a humanized anti-DLK1 monoclonal IgG1 kappa isotype antibody, site-specifically conjugated using Glycoconnect™[19] technology to the drug-linker PL1601, which contains HydraSpace™[20], a valine-alanine cleavable linker and the pyrrolobenzodiazepine (PBD) dimer SG3199 at a drug-to-antibody ratio of approximately 1.8 (Fig. 2A). HuBA-1-3D was derived from the murine monoclonal antibody (mAb) BA-1-3D after it was humanized by grafting the complementary determining region (CDR) of murine BA-1-3D into human IgG1 frameworks. Binding of HuBA-1-3D and ADCT-701 to human and cynomolgus monkey DLK1 showed similar affinities using surface plasma resonance analysis, and no binding affinity was lost in ADCT-701 after conjugating HuBA-1-3D to the drug-linker PL1601[10]. The cytotoxic PBD payload of ADCT-701, SG3199, causes potent, cytotoxic DNA interstrand cross-linking of the minor groove of DNA[21]. We determined the in vitro cytotoxicity of ADCT-701 using three established ACC cell lines[22,23] with varying levels of DLK1 surface expression (Fig. 2B, C). Compared to the isotype-control ADC (B12-PL1601), ADCT-701 exhibited cytotoxicity in DLK1⁺ CU-ACC1 and H295R cells, but not in DLK1⁻ CU-ACC2 cells (Fig. 2D). However, DLK1⁻ CU-ACC2 cells, similar to CU-ACC1 and H295R cells, were sensitive to the PBD payload of ADCT-701 (Supplementary Fig. 2A). We next used CRISPR-Cas9 gene editing in the CU-ACC1 cell line and established multiple single cell clones with loss of surface DLK1 expression (Supplementary Fig. 2B). In CU-ACC1 *DLK1* KO clones 9 and 10 with complete loss of surface DLK1 expression, ADCT-701 cytotoxicity was abrogated (Supplementary Fig. 2C) thereby validating the DLK1-specific cytotoxicity of ADCT-701. We observed a similar lack of ADCT-701 cytotoxicity in the DLK1⁻ PCPG cell line hPheo1[24] (Supplementary Fig. 2D, E). Taken together, these data demonstrate that ADCT-701 exhibits in vitro cytotoxic activity in a DLK1-dependent manner.

We next evaluated the mechanism by which ADCT-701 induces cell death. Cellular internalization of an ADC after binding to a surface target is an essential step for ADC cytotoxicity[1]. To demonstrate that our anti-DLK1 mAb could be internalized efficiently, we quantified the cellular internalization rates of DLK1 antibody across ACC cell lines with varying DLK1 expression levels using imaging flow cytometry. DLK1 antibody was rapidly internalized in DLK1⁺ CU-ACC1 and H295R cells but not in DLK1⁻ CU-ACC2 cells (Fig. 2E). Next, consistent with previous studies demonstrating PBD induced DNA interstrand cross-links results in cell cycle arrest[25], we found that CU-ACC1 and H295R cells treated with ADCT-701 were blocked in the G2/M phase (Supplementary Fig. 3A and Supplementary Fig. 4A). We then determined the presence of DNA double-strand breaks by γH2AX, as well as apoptosis by cleaved caspase-3 and cleaved poly (adenosine diphosphate-ribose) polymerase (PARP). γH2AX, cleaved caspase 3, and cleaved PARP were upregulated in CU-ACC1 and H295R cells after ADCT-701, but not after B12-PL1601 treatment (Supplementary Fig. 3B). ADCT-701 treatment also significantly increased apoptosis (Annexin V + /PI + ) compared to untreated and B12-PL1601 treated cells (Supplementary Fig. 3C and Supplementary Fig. 4B). Collectively, these results suggest that ADCT-701 treatment induces cell death through DNA double-strand breaks, G2/M arrest, and ultimately apoptosis.

Due to the heterogeneous expression of DLK1 in ACC[17] (Fig. 1E), we next assessed for potential bystander killing by which hydrophobic payloads such as PBD can diffuse from target antigen-expressing cancer cells after direct ADC cytotoxicity into neighboring antigen-negative cancer cells[1]. Using a system in which *DLK1* KO CU-ACC1 cells were cultured with DLK1-expressing parental CU-ACC1 cells at various ratios, we observed greater cytotoxicity of *DLK1* KO CU-ACC1 cells than

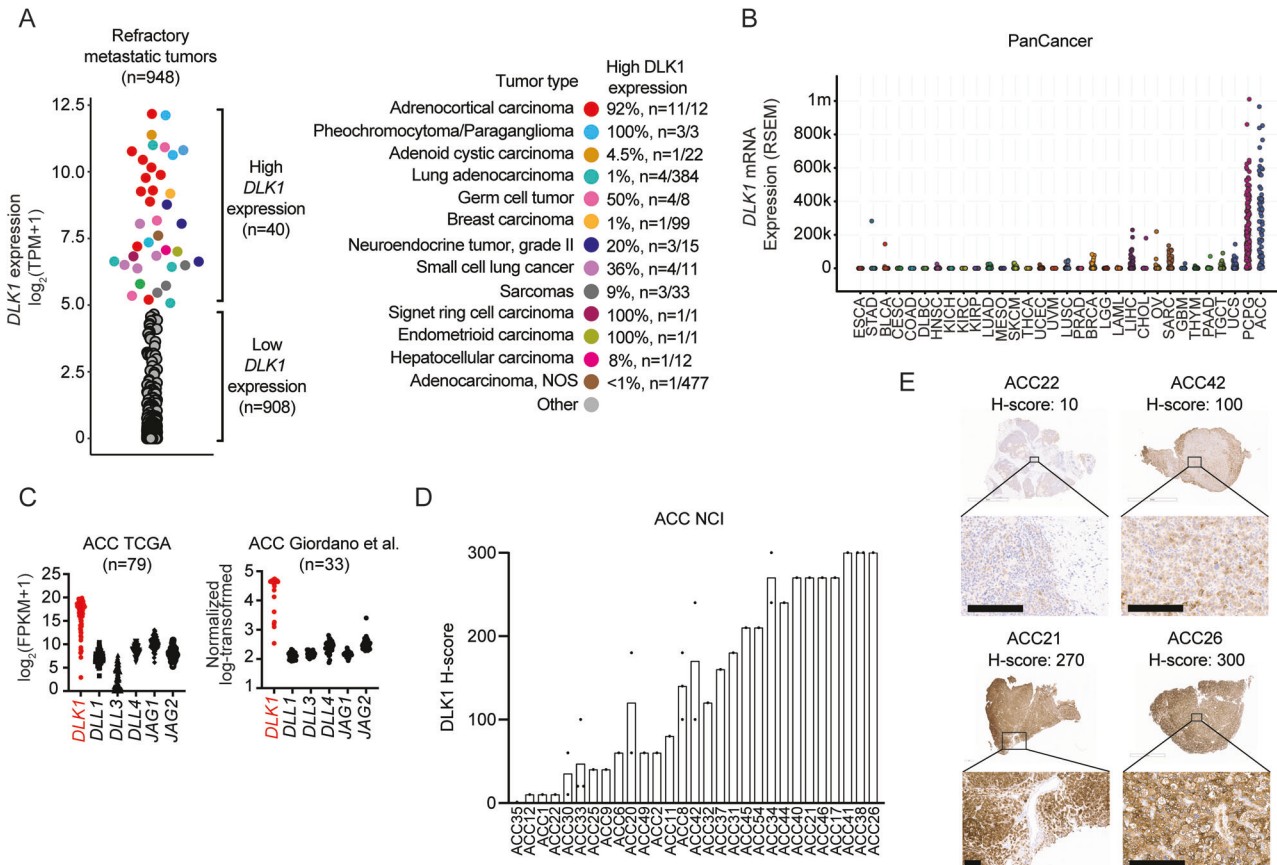

**Fig. 1 | Identification of DLK1 as the most highly expressed Notch ligand in adrenocortical carcinoma. A** *DLK1* mRNA expression across adult refractory metastatic cancers (*n* = 948). Tumor types with high *DLK1* expression are highlighted in the colors shown. The percentage of each tumor type with high *DLK1* expression is shown on the right. **B** *DLK1* mRNA expression in the TCGA PanCancer dataset. **C** Expression of Notch ligands from two independent ACC bulk RNA-seq datasets. **D** Quantification of DLK1 IHC staining among ACC NCI tumors (*n* = 38). Vertical bars with more than one tumor represent the mean H-score. **E** IHC images from four representative ACC NCI tumors with varying levels of DLK1 expression. Scale bars represent 200 μm. IHC immunohistochemistry. Source data are provided as a Source Data file.

expected and no cytotoxicity was observed with B12-PL1601 treatment (Supplementary Fig. 3D). We further investigated bystander killing by conditioned media transfer experiments in which DLK1⁺ CU-ACC1 cells were treated with ADCT-701 or B12-PL1601 before transferring the media to DLK1⁺ CU-ACC1 cells or *DLK1* KO CU-ACC1 cells. ADCT-701 conditioned media induced cytotoxicity in DLK1⁺ CU-ACC1 cells similar to treatment with ADCT-701 (Supplementary Fig. 3E). ADCT-701 conditioned media also elicited bystander killing, as demonstrated by greater cytotoxicity in *DLK1* KO CU-ACC1 cells compared with B12-PL1601 conditioned media or with ADCT-701 treatment (Supplementary Fig. 3E). Overall, these results indicate that ADCT-701 not only can target DLK1⁺ cells but can also indirectly induce cytotoxicity in DLK1⁻ cells through bystander killing.

### ADCT-701 has in vitro activity in DLK1⁺ ACC patient-derived organoids and induces anti-tumor responses in ACC cell line-derived and patient-derived xenografts

Since ACC is a rare cancer type with few available human cell lines[22,23], we sought to validate the in vitro cytotoxicity of ADCT-701 in a newly developed cohort of ACC short-term patient-derived organoids (PDOs) (defined as less than 5 total passages) (Supplementary Data 1). Overall, we found 50% (*n* = 6/12) of PDOs responded to ADCT-701 (Fig. 2F) and 50% (*n* = 6/12) of PDOs had no response (Fig. 2G). As expected, all ADCT-701 responders were DLK1⁺ based on a rightward shift of the anti-DLK1 flow cytometry histograms compared to unstained controls (Fig. 2H). However, among ADCT-701 non-responders, only 50% (*n* = 3/6) were DLK1⁻ (Fig. 2I), suggesting the lack of

target antigen expression may not be a mechanism of resistance to ADCT-701.

Next, to further explore the potential for targeting DLK1 in ACC, we evaluated responses to ADCT-701 (1 mg/kg as a single dose[18]) among DLK1⁺ human ACC cell line-derived xenograft and ACC patient-derived xenograft (PDX) models (Fig. 3A, B). ADCT-701 treatment elicited durable anti-tumor responses and significantly prolonged the survival of both CU-ACC1 and H295R tumor-bearing mice compared with tumor-bearing mice treated with saline or B12-PL1601 (Fig. 3A and Supplementary Fig. 5A). However, H295R tumors eventually became resistant to ADCT-701, whereas CU-ACC1 tumors remained sensitive to ADCT-701 for up to 100 days. While the antibody within ADCT-701 targets human DLK1, not mouse Dlk1, no body weight loss in mice was observed with ADCT-701 treatment (Supplementary Fig. 5B), suggesting minimal off-target payload activity.

We next investigated the anti-tumor activity of ADCT-701 among three DLK1⁺ ACC PDX models: 164165, 592788, and POBNCI_ACC004 (Fig. 3B and Supplementary Fig. 6A). All three PDX models were validated as ACC tumors based on IHC expression of the common NE marker synaptophysin, the adrenal-specific marker SF1, and the cell proliferation marker Ki67 (Supplementary Fig. 6B–D). ADCT-701 treatment led to tumor growth inhibition and significant lengthening of survival of 164165 PDX and 592788 PDX mice (Fig. 3B and Supplementary Fig. 5C). Strikingly, ADCT-701 induced complete responses in all treated POBNCI_ACC004 PDX tumors (5/5). However, 3/5 POBNCI_ACC004 PDX tumors quickly recurred and did not respond to ADCT-701 re-treatment (Fig. 3B). Similar to treated xenografts,

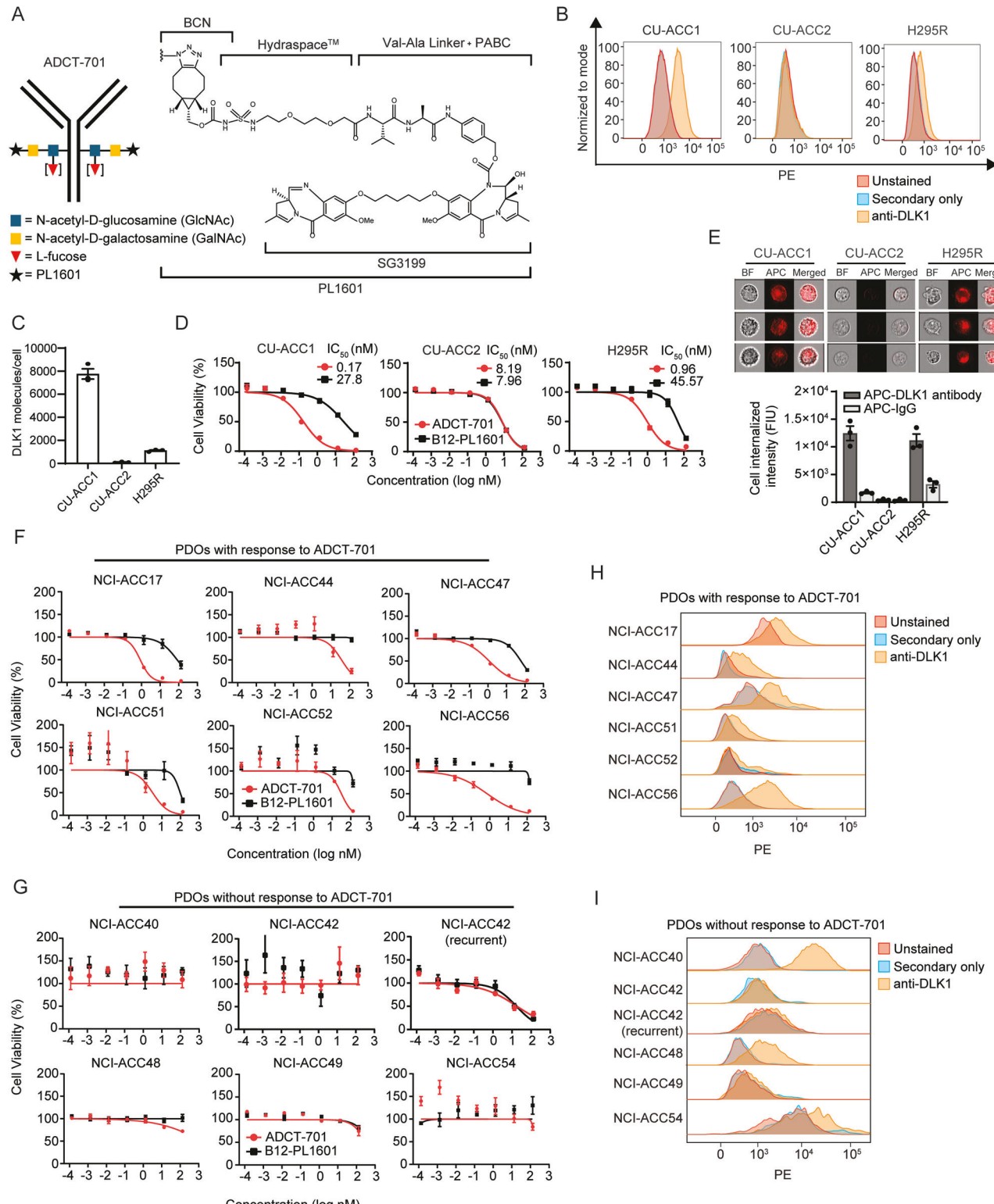

**Fig. 2 | ADCT-701, a DLK1 targeting antibody-drug conjugate, has potent in vitro activity in adrenocortical carcinoma. A** Schematic structure of ADCT-701, a DLK1 targeting antibody drug conjugate. **B** Representative surface expression of DLK1 among ACC cell lines: CU-ACC1, CU-ACC2, and H295R. **C** DLK1 molecules/cell in ACC cell lines (n = 3 independent experiments). **D** ADCT-701 cytotoxicity among CU-ACC1, CU-ACC2, and H295R cells. Cells were treated with ADCT-701 and B12-PL1601 (non-targeted control ADC) for 7 days (data representative of n = 3 independent experiments). **E** Representative imaging flow cytometry images and signal intensity analysis (n = 3 biological replicates) showing cellular internalization of DLK1 antibodies in CU-ACC1, CU-ACC2, and H295R cells. **F** Cytotoxic activity of ADCT-701 responsive (n = 6) and (**G**) ADCT-701 non-responsive (n = 6) ACC short-term patient-derived organoids (PDOs). Cells were treated with ADCT-701 and B12-PL1601 for 7 days. Data representative of n = 1 (ACC17, ACC44, ACC56, and ACC49) or n = 2 (ACC47, ACC51, ACC52, ACC40, ACC42, ACC48, and ACC54) independent experiments. Flow cytometry histograms assessing DLK1 among ADCT-701 **H** responsive and **I** non-responsive ACC PDOs. Error bars represent mean values ± S.E.M. Drug response curve data are presented as mean values ± S.E.M. Source data are provided as a Source Data file.

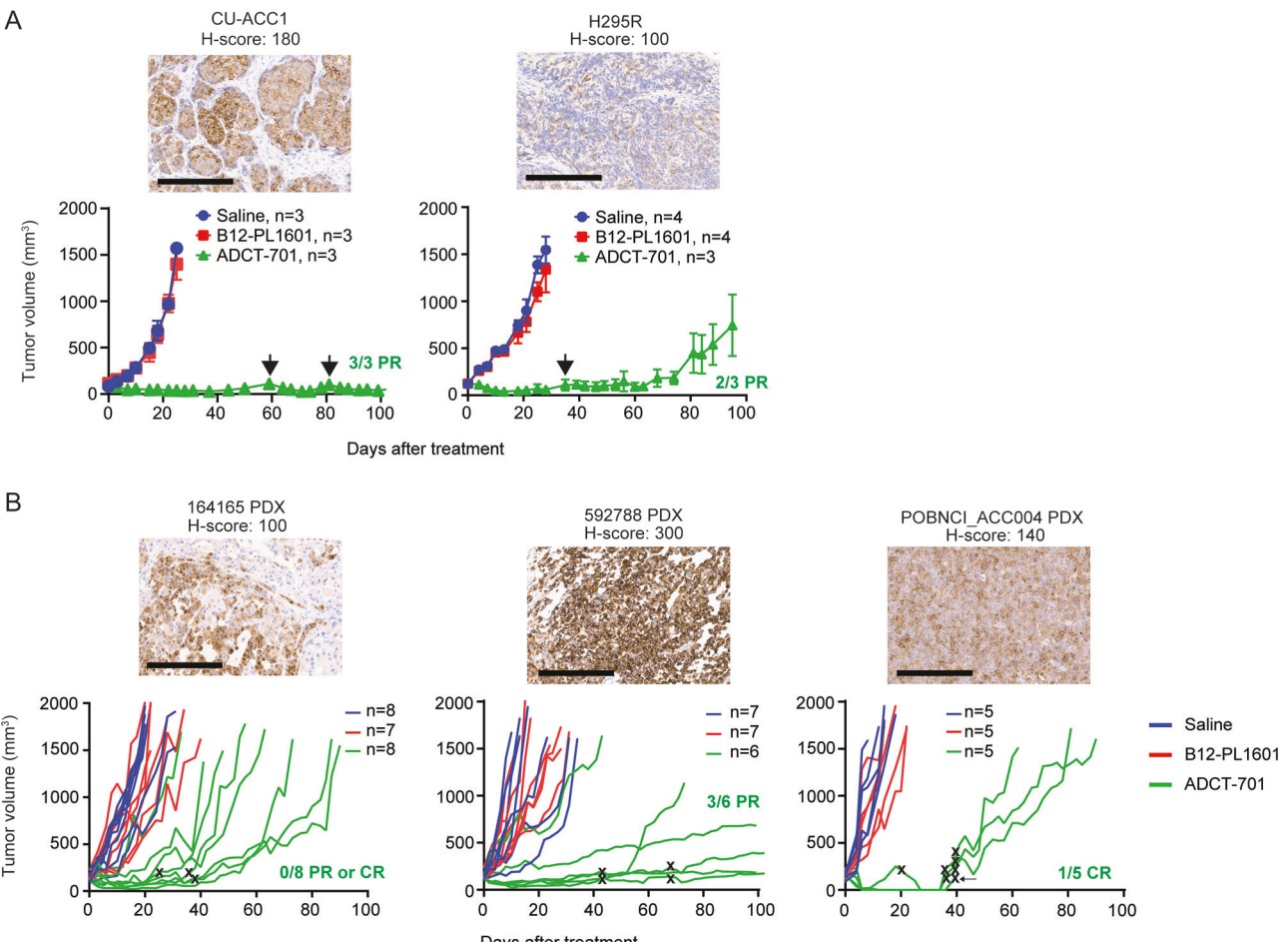

**Fig. 3 | ADCT-701 induces robust anti-tumor responses in DLK1⁺ ACC tumors.**
**A** CU-ACC1 and H295R xenograft tumor growth curves after treatment with saline
(CU-ACC1: $n = 3$; H295R: $n = 4$), B12-PL1601 (CU-ACC1: $n = 3$; H295R: $n = 4$), or ADCT-
701 (CU-ACC1: $n = 3$; H295R: $n = 3$) (1 mg/kg) (initial tumor size reached an average of
100–150 mm³). Additional doses of ADCT-701 indicated by arrows. Error bars
represent mean values ± S.E.M. **B** ACC PDXs 164165, 592788, and POBNCI_ACC004
tumor growth curves after treatment with saline (164165: $n = 8$; 592788: $n = 7$;
POBNCI_ACC004: $n = 5$), B12-PL1601 (164165: $n = 7$; 592788: $n = 7$; POBNCI_ACC004:

$n = 5$) or ADCT-701 (164165: $n = 8$; 592788: $n = 6$; POBNCI_ACC004: $n = 5$) (1 mg/kg)
(initial tumor size reached an average of 100–200 mm³). Each line shown represents
an individual mouse within a given experiment. The X symbol indicates the admin-
istration of ADCT-701 re-dosing. Arrow indicates unexpected death of 1 POBN-
CI_ACC004 tumor-bearing mouse prior to endpoint. DLK1 immunohistochemistry
with H-scores shown above each individual xenograft or PDX tumor. PR partial
response, CR complete response. Scale bars represent 200 μm. Source data are
provided as a Source Data file.

ADCT-701 was well-tolerated in PDX models as determined by body
weight measurements (Supplementary Fig. 5D).

## ABCB1, a drug efflux protein, mediates intrinsic and acquired ADC and chemotherapy resistance among DLK1⁺ ACC pre-clinical models

We next sought to decipher mechanisms of resistance to ADCT-701 in
our DLK1⁺ ACC pre-clinical models. As payload insensitivity is known to
mediate ADC resistance[1], we first tested the cytotoxicity of PBD across
DLK1⁺ ADCT-701 responder and non-responder PDOs. Strikingly, we
observed extreme resistance to PBD among non-responder PDOs, with
PBD average IC50s of 73 nM in NCI-ACC40, 31 nM in NCI-ACC48, and
27 nM in NCI-ACC54, which represents close to 1000x greater resis-
tance than previously reported for PBD[26] (Fig. 4A). We then explored
the activity of common therapies used to treat ACC in the clinic[14]
(mitotane, etoposide, doxorubicin, and carboplatin) among DLK1⁺
ACC PDOs with and without response to ADCT-701 and PBD. We
observed resistance to chemotherapy in several DLK1⁺ ACC PDOs
without response to ADCT-701 and PBD (Supplementary Fig. 7A). We
next looked for potential mechanisms of resistance to PBD by ana-
lyzing our PDO models for expression of commonly upregulated drug
efflux pumps of the ABC transporter family[26]. *ABCB1* and *ABCG2* had

the greatest difference in expression between DLK1⁺ ADCT-701
responder PDOs (NCI-ACC44, NCI-ACC51, and NCI-ACC56) and DLK1⁺
ADCT-701 non-responder PDOs (NCI-ACC40, NCI-ACC48, and NCI-
ACC54) (Fig. 4B), suggesting these drug efflux genes could explain the
difference in PBD resistance in corresponding PDOs. Indeed, the DLK1⁺
ADCT-701 non-responder PDOs had much higher surface expression of
ABCB1 than the DLK1⁺ ADCT-701 responder PDOs (Fig. 4C). Further-
more, co-treatment of 3 different ABCB1 inhibitors (valspodar, elacri-
dar, and tariquidar) with ADCT-701 and PBD in NCI-ACC40 and NCI-
ACC48 PDOs showed dramatic reversal of resistance (Fig. 4D and
Supplementary Fig. 7B). ABCB1 inhibitors more modestly increased
sensitivity to ADCT-701 and PBD in the NCI-ACC51 PDO demonstrating
a functionally lower level of ABCB1 activity in this model compared to
both NCI-ACC40 and NCI-ACC48 PDOs (Supplementary Fig. 7C).
ABCB1 inhibition only minimally affected cell growth in these three
PDOs (Supplementary Fig. 7D). Thus, these results indicate that pri-
mary in vitro resistance to ADCT-701 can be mediated by high
expression and activity of the drug efflux protein ABCB1.

We next sought to determine if ABCB1 expression and activity
could also explain differences in ADCT-701 in vivo activity. Among our
ACC cell-line derived xenografts, CU-ACC1 had lower surface ABCB1
expression than H295R (Supplementary Fig. 8A), which could at least

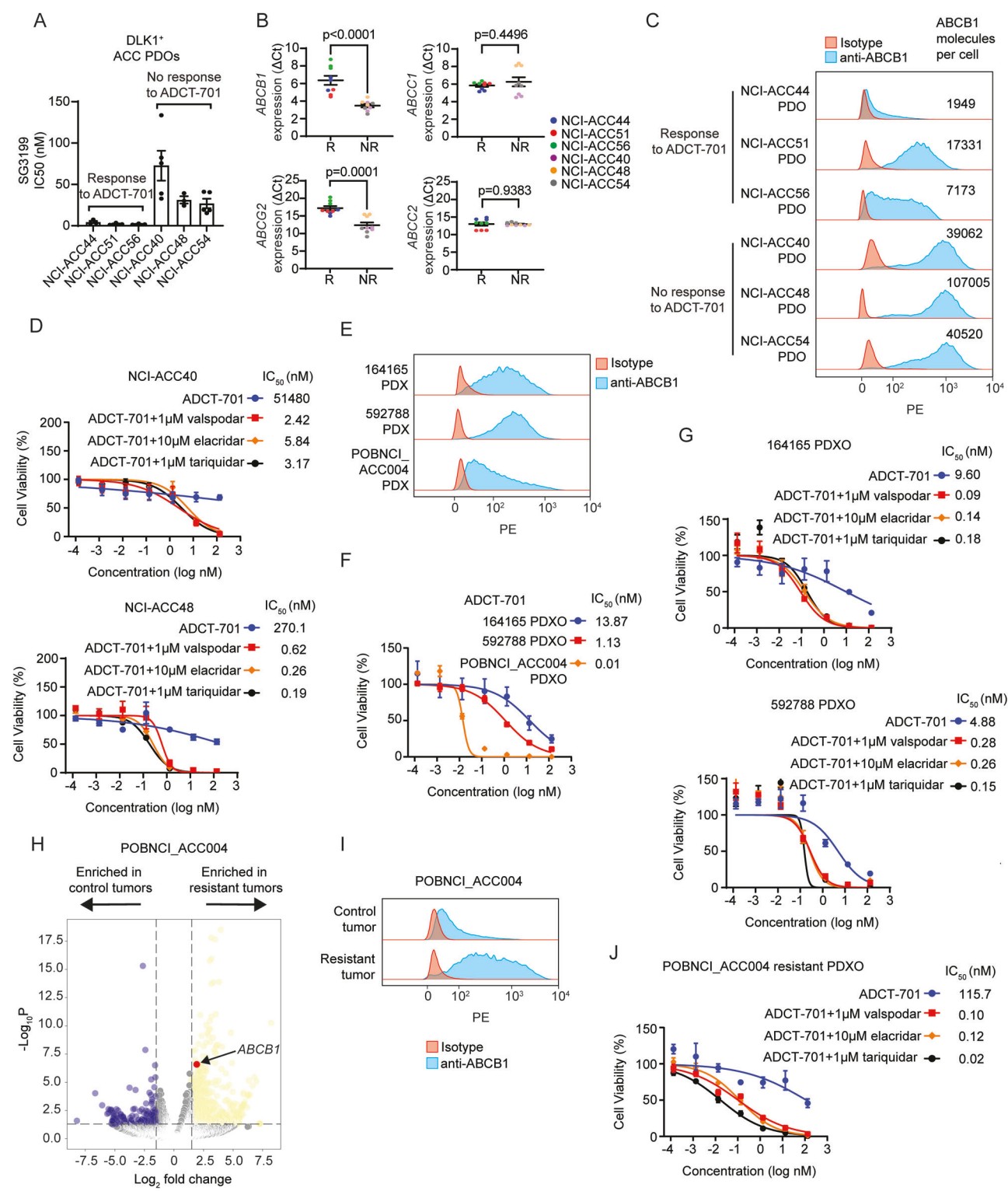

partially explain the long-term tumor control with ADCT-701 treatment in CU-ACC1 but not H295R tumors (Fig. 3A). Among our ACC PDXs, there was considerably lower surface ABCB1 expression in POBNCI_ACC004 (Fig. 4E), which had initial complete responses with ADCT-701 treatment, compared to both 164165 and 592788 (Fig. 3B), which had partial but no complete responses with ADCT-701 treatment. To further test the role of ABCB1 in relation to ADCT-701 activity in these models, we developed PDX-derived organoids from untreated POBNCI_ACC004, 164165 and 592788 PDX tumors (i.e., PDXOs). Consistent with our in vivo data, the POBNCI_ACC004 PDXO was much

more sensitive to ADCT-701 and PBD than 164165 and 592788 PDXOs (Fig. 4F and Supplementary Fig. 8B). ABCB1 inhibitors also increased sensitivity to ADCT-701 and PBD among 164165 and 592788 PDXOs (Fig. 4G and Supplementary Fig. 8C), demonstrating that ABCB1 drug efflux activity is a mechanism of intrinsic resistance to ADCT-701 in these two models.

Although POBNCI_ACC004 PDX tumors had initial complete responses to ADCT-701 treatment, these tumors quickly relapsed and were unresponsive to additional ADCT-701 doses (Fig. 3B). Therefore, to assess mechanisms of acquired resistance to ADCT-701, we

**Fig. 4 | ABCB1, a drug efflux protein, mediates intrinsic and acquired resistance to ADCT-701. A** Cytotoxicity of SG3199 among ADCT-701 responsive (NCI-ACC44, NCI-ACC51, and NCI-ACC56) and non-responsive (NCI-ACC40, NCI-ACC48, and NCI-ACC54) DLK1⁺ ACC short-term patient-derived organoids (PDOs). Cells were treated with SG3199 for 3 days (n = 3 independent experiments for ACC44, ACC51, ACC56, and ACC48 PDOs; n = 5 independent experiments for ACC40 and ACC54 PDOs). **B** Drug transporter mRNA expression in DLK1⁺ ADCT-701 responder and non-responder PDOs as measured by quantitative RT-PCR. Results were presented as Δcycle threshold (ΔCt). Each color represents an individual PDO. R responder, NR non-responder (n = 3 independent experiments). Unpaired t tests were used to calculate two-tailed p-values. **C** Flow cytometry histograms assessing ABCB1 and the number of ABCB1 molecules per cell of DLK1⁺ ADCT-701 responsive (NCI-ACC44, NCI-ACC51, and NCI-ACC56) and non-responsive (NCI-ACC40, NCI-ACC48, and NCI-ACC54) ACC PDOs. **D** ADCT-701 cytotoxicity in the NCI-ACC40 and NCI-ACC48 PDOs with and without treatment with ABCB1 inhibitors (1 μM valspodar, 10 μM elacridar and 1 μM tariquidar). Cells were treated with ADCT-701 combined with or without ABCB1 inhibitors for 7 days (data representative of n = 3 independent experiments). **E** Flow cytometry histograms assessing ABCB1 among 164165,

592788 and POBNCI_ACC004 PDXs (data representative of n = 2 independent experiments). **F** ADCT-701 cytotoxicity in 164165, 592788 and POBNCI_ACC004 PDX-derived organoids. Cells were treated with ADCT-701 for 7 days (data representative of n = 3 independent experiments). **G** ADCT-701 cytotoxicity in the 164165 and 592788 PDX-derived organoids treated with or without ABCB1 inhibitors (1 μM valspodar, 10 μM elacridar and 1 μM tariquidar). Cells were treated with ADCT-701 combined with or without ABCB1 inhibitors for 7 days (data representative of n = 3 independent experiments). **H** Volcano plot of differentially expressed genes in control tumors (n = 4) versus post-ADCT-701 acquired resistant tumors (n = 3) in POBNCI_ACC004 PDX. Wald test negative log₁₀ p-values are shown on the y-axis. **I** Flow cytometry histograms assessing ABCB1 among ADCT-701 resistant and control POBNCI_ACC004 PDX tumors. **J** ADCT-701 cytotoxicity in the ADCT-701 resistant POBNCI_ACC004 PDX-derived organoid treated with or without ABCB1 inhibitors (1 μM valspodar, 10 μM elacridar and 1 μM tariquidar). Cells were treated with ADCT-701 combined with or without ABCB1 inhibitors for 7 days (data representative of n = 3 independent experiments). Error bars represent mean values ± S.E.M. Source data are provided as a Source Data file.

performed RNA-seq on POBNCI_ACC004 PDX tumors resistant to ADCT-701 treatment (n = 3) and saline treated control tumors (n = 4). Using differential gene expression analysis, we observed significant upregulation of *ABCB1* expression in POBNCI_ACC004 PDX tumors resistant to ADCT-701 compared to controls (Fig. 4H and Supplementary Data 2). Surface ABCB1 expression was also highly upregulated in resistant compared to untreated POBNCI_ACC004 tumors (Fig. 4I). We then developed a PDXO from a POBNCI_ACC004 resistant tumor and found that ABCB1 inhibitors re-sensitized the POBNCI_ACC004 resistant PDXO to both ADCT-701 and PBD (Fig. 4J and Supplementary Fig. 8C), demonstrating the role of this drug efflux transporter in mediating acquired resistance to ADCT-701. ABCB1 inhibition alone had a minimal effect on cell growth in 164165, 592788, and POBNCI_ACC004 resistant PDXOs (Supplementary Fig. 8D). Lastly, unlike in neuroblastoma[10], we found no difference in DLK1 expression by IHC between pre- and post- ADCT-701 treated ACC PDX tumors (Supplementary Fig. 8E) suggesting that selection and outgrowth of DLK1 negative cells does not contribute to ADCT-701 acquired resistance in ACC.

## ADCT-701 elicits complete, durable responses in DLK1⁺ small cell lung cancer tumors without ABCB1 expression

As we found *DLK1* to be expressed in a subset of metastatic cancers apart from ACC (Fig. 1A), we hypothesized that ADCT-701 would be highly effective against DLK1⁺ tumors with low or no ABCB1 expression, such as SCLC. We therefore screened SCLC cell lines for expression of DLK1 and found 22% (n = 11/51) were DLK1⁺ (Supplementary Fig. 9A). We then selected three DLK1⁺ SCLC cell lines (H524, H146, and H1436) and confirmed that cell surface DLK1 expression was at a level equal to or higher than the known SCLC target DLL3 (Fig. 5A, B). All three SCLC cell lines also lacked surface ABCB1 expression (Fig. 5C) and were highly sensitive to both PBD and ADCT-701 in vitro (Fig. 5D, E). In vivo, ADCT-701 treatment resulted in complete responses and long-term tumor-free survival compared to B12-PL1601 and saline in all three SCLC xenograft models (Fig. 5F and Supplementary Fig. 9B) without appreciable body weight loss (Supplementary Fig. 9C). Notably, the SCLC H146 xenograft, which had very low DLK1 expression (H-score 30), also had long-term complete responses with ADCT-701 treatment (Fig. 5F). Thus, our results indicate that ADCT-701 can effectively target DLK1⁺ tumors with low or no ABCB1 expression.

## DLK1 is a major regulator of ABCB1 and chemoresistance in ACC

We next sought to assess whether DLK1 has a functional role in ACC using a subset our established CU-ACC1 *DLK1* KO cells, which had no expression of both full length and cleaved DLK1 (clones 9 and 10)

(Fig. 6A). As DLK1 has been shown to either activate or inhibit NOTCH1 signaling in different model systems[27], we assessed expression of NOTCH1 in *DLK1* KO clones compared to *DLK1* WT parental cells and observed upregulation of total NOTCH1 and the active, intracellular domain (ICD) of NOTCH1 (N1ICD) (Fig. 6B). In *DLK1* KO clones compared to parental cells, there was also a dramatic reduction in the NE protein synaptophysin (Fig. 6B) consistent with the known role of active Notch signaling in downregulating NE gene expression[28]. Correspondingly, we observed a significant negative correlation between *NOTCH1* and *DLK1* expression among TCGA ACC tumors (Fig. 6C) as well as in SCLC primary tumors[29] (Supplementary Fig. 9D). *DLK1* and *NOTCH1* expression were also significantly higher and lower, respectively, in ACC tumors compared to the normal adrenal gland (Fig. 6D).

Given that prior work has shown Notch-active tumors with low NE gene expression (i.e., non-NE) to be chemoresistant[30], we performed cytotoxicity assays with PBD in CU-ACC1 cells with and without *DLK1* KO. Surprisingly, *DLK1* KO clones were much more sensitive to PBD (Fig. 6E) and chemotherapeutics such as etoposide and doxorubicin than CU-ACC1 parental cells (Supplementary Fig. 10A). Strikingly, we observed near complete loss of ABCB1 surface expression in *DLK1* KO clones compared to parental cells, which had a broad, bi-modal distribution of ABCB1 (Fig. 6F). In contrast, siRNA downregulation of DLK1 showed minimal changes in NOTCH1, N1ICD, and ABCB1 expression (Supplementary Fig. 10B, C) suggesting complete loss of DLK1 is required for NOTCH1 activation and ABCB1 downregulation. As we observed *DLK1* KO cells to be completely adherent compared to a mixed phenotype (suspension and adherent) of CU-ACC1 parental cells (Supplementary Fig. 10D), we isolated suspension and adherent CU-ACC1 cells (Supplementary Fig. 10E) and found suspension CU-ACC1 cells to have high expression of DLK1 and SYP and low expression of N1ICD (Supplementary Fig. 10F). In contrast, adherent CU-ACC1 cells showed low expression of DLK1 and SYP and high expression of N1ICD (Supplementary Fig. 10F). CU-ACC1 adherent cells also had lower expression of surface ABCB1 with modest differences in chemosensitivity compared to CU-ACC1 suspension cells (Supplementary Fig. 10G, H).

To validate our *DLK1* KO results, we assessed expression of NOTCH1 and NE proteins in ACC PDOs with high DLK1 expression (NCI-ACC40 and NCI-ACC48) and an ACC PDO with no DLK1 expression (NCI-ACC49) (Fig. 6G). We observed higher expression of N1ICD and much lower expression of SYP in the DLK1⁻ NCI-ACC49 PDO compared to the DLK1⁺ NCI-ACC40 and NCI-ACC48 PDOs (Fig. 6G). The NCI-ACC49 PDO was also highly sensitive to PBD, etoposide and doxorubicin (Supplementary Fig. 10I) and surface ABCB1 was not expressed (Fig. 6H). We next overexpressed *N1ICD* in CU-ACC1 cells, and similar to the *DLK1* KO clones, we observed decreased expression of SYP (Fig. 6I) and near complete downregulation of surface ABCB1 (Fig. 6J).

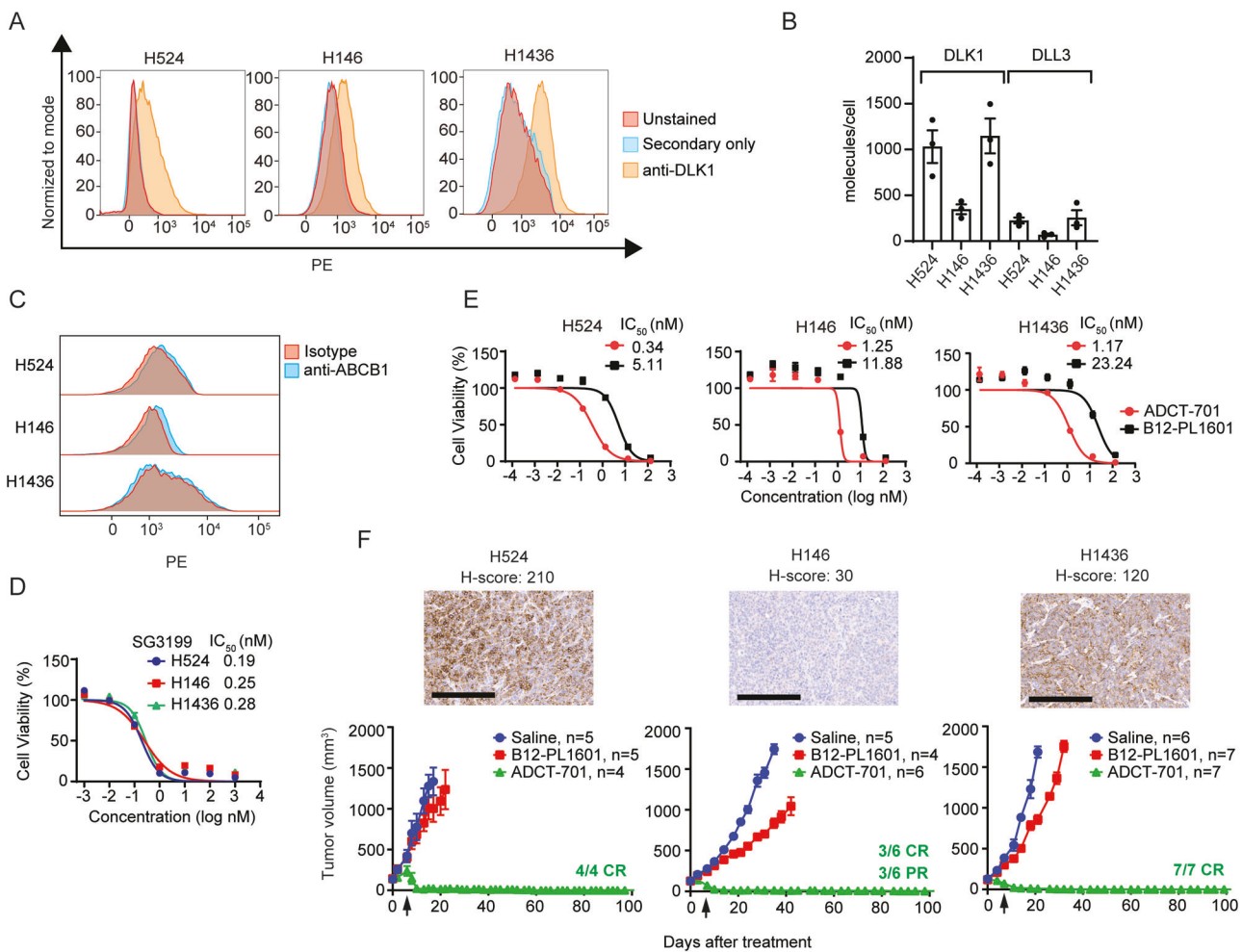

**Fig. 5 | ADCT-701 elicits complete, durable responses in DLK1⁺ small cell lung cancer tumors without ABCB1 expression. A** Cell surface DLK1 expression by flow cytometry in 3 small cell lung cancer (SCLC) cell lines (H524, H146, and H1436) (data representative of $n = 3$ independent experiments). **B** DLK1 molecules/cell relative to DLL3 among SCLC cell lines ($n = 3$ independent experiments). Error bars represent mean values ± S.E.M. **C** Flow cytometry histograms assessing ABCB1 in SCLC cell lines (data representative of $n = 3$ independent experiments). **D** SG3199 cytotoxicity in SCLC cell lines. Cells were treated with SG3199 for 3 days (data representative of $n = 4$ independent experiments). **E** ADCT-701 cytotoxicity among

SCLC cell lines. Cells were treated with ADCT-701 or B12-PL1601 for 7 days (data representative of $n = 4$ independent experiments). **F** Tumor growth curves of SCLC xenograft models after treatment (1 mg/kg) with saline (H524: $n = 5$; H146: $n = 5$; H1436: $n = 6$), B12-PL1601 (H524: $n = 5$; H146: $n = 4$; H1436: $n = 7$) or ADCT-701 (H524: $n = 4$; H146: $n = 6$; H1436: $n = 7$) when tumor size reached an average of 100–150 mm³. Arrows indicate re-treatment with ADCT-701. Scale bars represent 200 μm. Error bars represent mean values ± S.E.M. Source data are provided as a Source Data file.

*N1ICD*-overexpressing CU-ACC1 cells were also more sensitive to PBD and etoposide than CU-ACC1 parental cells (Supplementary Fig. 10J). Lastly, we analyzed the relationship between *NOTCH1* and *ABCB1* expression in bulk and single-cell RNA-seq datasets. Across TCGA ACC tumors and normal adrenals, we observed a significant negative correlation between *NOTCH1* and *ABCB1* expression (Fig. 6K, L). Correspondingly, single-cell RNA-seq data from 18 ACC metastatic tumors (*Aber* et al. *manuscript in submission*) demonstrated significantly lower *ABCB1* expression among cells with high compared to low *NOTCH1* expression (Fig. 6M). Altogether, our data suggest a model by which DLK1, through inhibition of NOTCH1 signaling, maintains high ABCB1 expression and imparts ADC and chemoresistance in ACC (Fig. 6N). Based on our data, DLK1-directed ADCs would also be expected to have greater activity in ACC tumors with positive but low DLK1 expression due to decreased ABCB1 expression (Fig. 6N).

**A first-in-human phase I clinical trial of ADCT-701 in patients with ACC and neuroendocrine neoplasms**
Based on the pre-clinical efficacy of ADCT-701 in ACC and SCLC, as well as parallel data in neuroblastoma[10], a first-in-human phase 1 clinical trial

was developed to test the safety and preliminary efficacy of ADCT-701 in adult patients with ACC and NE neoplasms. This trial (NCT06041516) is currently recruiting patients with the primary objective to determine the maximum tolerated dose (MTD) of ADCT-701.

## Discussion
In this work, we have identified DLK1 as a cancer cell surface antigen that can be successfully targeted with an ADC in pre-clinical models of refractory metastatic cancers, namely ACC and SCLC. While ADCs are an effective and increasingly common cancer therapy, approval is currently limited to select malignancies (i.e., breast cancer, urothelial cancers, and ovarian cancers) with overall few antigen targets (i.e., TROP2, nectin-4, HER2, tissue factor, and folate receptor alpha[1]). Thus, identifying new and optimal cell surface targets, such as DLK1, is an important step towards broadening the therapeutic potential of ADCs. Indeed, based on our pre-clinical data, we have initiated a first-in-human phase 1 clinical trial with an ADC targeting DLK1 (NCT06041516). To our knowledge, this is the first ADC clinical trial for patients with ACC and NE neoplasms, including rare NE malignancies such as PCPG and adult neuroblastoma.

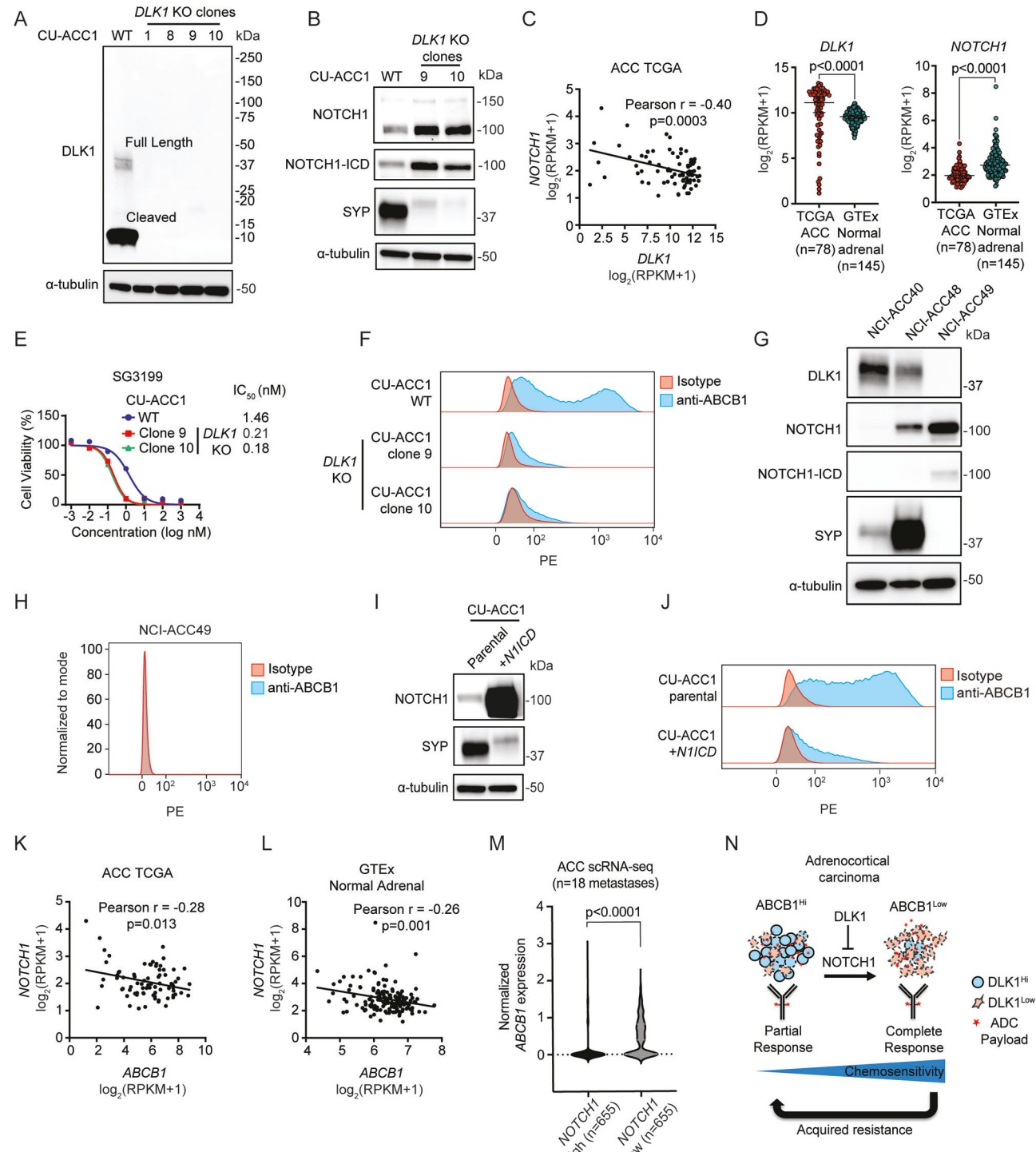

**Fig. 6 | DLK1 is a major regulator of ABCB1 and chemoresistance in adreno-cortical carcinoma. A** Immunoblot analysis of DLK1 and loading control (α-tubulin) proteins in CU-ACC1 cells with and without *DLK1* KO. Four single-cell KO clones are shown. **B** Immunoblot analysis of NOTCH1 signaling, total NOTCH1 and NOTCH1 intracellular domain (ICD), NE marker synaptophysin (SYP), and loading control (α-tubulin) proteins with and without *DLK1* KO in CU-ACC1 cells. Two single-cell KO clones are shown. **C** Correlation between *NOTCH1* and *DLK1* expression among TCGA ACC tumors. Pearson correlation coefficients with two-tailed p-values are shown. **D** *DLK1* and *NOTCH1* expression in TCGA ACC tumors and normal adrenals. Unpaired t tests were used to calculate two-tailed *p*-values. Error bar represents mean values ± 95% C.I. **E** SG3199 cytotoxicity in CU-ACC1 parental and *DLK1* KO clones. Cells were treated with SG3199 for 3 days (data representative of *n* = 4 independent experiments). Error bars represent mean values ± S.E.M. **F** Flow cytometry histograms assessing ABCB1 in CU-ACC1 cells with

and without *DLK1* KO. **G** Immunoblot analysis of DLK1, total NOTCH1 and NOTCH1-ICD, SYP, and α-tubulin proteins in DLK1⁺ NCI-ACC40, DLK1⁺ NCI-ACC48 and DLK1⁻ ACC49 PDOs. **H** Flow cytometry histograms assessing ABCB1 in DLK1 negative NCI-ACC49 PDOs. **I** Immunoblot analysis of total NOTCH1 (to detect the NOTCH1-ICD plasmid expression), SYP, and α-tubulin proteins in CU-ACC1 cells with and without *NOTCH1-ICD* overexpression. **J** Flow cytometry histograms assessing ABCB1 in CU-ACC1 cells with and without *N1ICD* overexpression. **K** Correlation between *NOTCH1* and *ABCB1* expression among TCGA ACC tumors and (**L**) among GTEx normal adrenal tissues. Pearson correlation coefficients with two-tailed p-values are shown. **M** Single cell RNA-seq data of *ABCB1* expression comparing high *NOTCH1* to low *NOTCH1* expressing cells from 18 ACC metastatic tumors. Unpaired t tests were used to calculate two-tailed *p*-values. **N** Model summarizing the findings of the current study. For immunoblots, experiments were performed two times with similar results. Source data are provided as a Source Data file.

In addition to identifying DLK1 as a new immunotherapeutic target, we demonstrate a new functional role for DLK1 in conferring ADC and chemoresistance. *DLK1* is a maternally imprinted, paternally expressed gene within the *DLK1-DIO3* gene cluster[31] that encodes a transmembrane glycoprotein with a structure similar to canonical Notch ligands. In normal tissues, DLK1 is known to be involved in multiple cellular differentiation processes such as adipogenesis[32], hematopoiesis and stem cell homeostasis[33], neurogenesis[34], angiogenesis[35], and muscle regeneration[36]. In the adrenal cortex, DLK1 is widely expressed during development but postnatally is confined to undifferentiated cortical progenitor cells[37]. In cancer, DLK1 inhibits differentiation but has been shown to have both pro- and anti-proliferative effects[27], likely reflecting the complex relationship between DLK1 and NOTCH1 signaling across disease models[38,39]. In ACC, while the role of DLK1 in tumorigenesis is still not well-defined, DLK1 expression is associated with worse recurrence-free survival[17]. In two ACC single-nuclei RNA-sequencing studies, *DLK1* was highly expressed in clusters with abnormal *DLK1* locus copy number states from primary surgical resected ACC tumors[40] and the Notch signaling pathway was suppressed across most cell clusters from both primary and metastatic ACC tumors[41]. Our data demonstrate that DLK1 regulates expression and activity of the multidrug efflux pump ABCB1 through NOTCH1 signaling, as we observed dramatic downregulation of surface ABCB1 in cells with either *DLK1* KO or *N1ICD* overexpression. We believe these data are particularly relevant to the clinic as chemotherapy resistance in ACC has long been attributed to high expression of ABCB1[2,42], which is known to be one of several genes highly expressed in the adrenal cortex[15]. Thus, inhibition of DLK1 could be a strategy to reduce resistance to chemotherapy in ACC, particularly the EDP (etoposide, doxorubicin, cisplatin) regimen, which includes two chemotherapeutic drugs known to be ABCB1 transport substrates (etoposide and doxorubicin). EDP chemotherapy, when combined with mitotane, is widely used to treat advanced ACC but does not extend overall survival compared to prior therapies[43].

Our experimental data also uncover a role for DLK1 in transdifferentiating cells from a NE to non-NE state, which, to our knowledge, has not been previously known, although DLK1 has been observed to be upregulated in NE tumors[29,44]. Interestingly, adrenocortical carcinomas are not typically categorized as NE tumors as they are not of neuroepithelial origin and they generally lack expression of NE genes such as chromogranin A[45]. Indeed, NE scoring systems have used gene expression data from the normal adrenal cortex to identify non-NE genes compared to NE genes in the normal adrenal medulla[46]. However, ACC tumors commonly express NE genes such as synaptophysin[47], and our data demonstrate that synaptophysin is regulated by DLK1 through NOTCH1. Correspondingly, synaptophysin was observed to be highly upregulated in high compared to low *DLK1* expressing ACC tumor regions based on spatial transcriptomic analyses[17]. Low or no expression of synaptophysin on routine ACC tumor specimens, albeit likely low in prevalence, could indicate a less NE-driven state (i.e., DLK1^low/NOTCH1^high/ABCB1^low) with sensitivity to chemotherapy or an ADC. Counterintuitively, our data suggest that high DLK1 expression may not be an optimal biomarker for a DLK1-directed ADC, as DLK1^high tumors would be expected to have high ABCB1 expression and thereby demonstrate payload resistance. Rather, tumors with low DLK1 expression (which are able to bind and internalize a DLK1-directed ADC) may exhibit the most optimal ADC response due to low ABCB1 expression.

The direct link we propose between DLK1, NOTCH1 signaling, and ABCB1 expression also suggests that ABCB1 inhibition could improve anti-tumor responses to DLK1-directed ADCs. However, a clinical trial testing the addition of the ABCB1 antagonist, tariquidar, to chemotherapy among ACC patients did not prolong survival[48]. Toxicity of the combination of ABCB1 inhibitors and chemotherapy (such as to bone marrow whose stem cells are protected from chemotherapy by expression of ABC efflux transporters such as ABCG2 and ABCB1)

should be much less of a concern when ABCB1 inhibitors are combined with targeted therapy such as ADCs, which have reduced off-target toxicity. Another strategy for future ADCs could be to use payloads that are not substrates for drug efflux transporters[49].

Beyond ADCs, degrader-antibody conjugates[1] targeting DLK1 may be a strategy to downregulate DLK1, which could potentially sensitize ACC tumors to chemotherapy or other ADCs. Moreover, there are now multiple immunotherapeutic strategies to target cancer cell surface proteins, such as CAR T cells and bi-specific T cell engagers (BiTEs). Indeed, DLK1-directed CAR T cells have been shown to have preclinical efficacy among DLK1-expressing hepatocellular carcinoma models[50]. CAR T cells may be a particularly attractive option in ACC given the high level of chemoresistance; however, CAR T cells generally have more toxicity than ADCs[51], and thus, it may be advisable to accrue safety information on targeting DLK1 from our phase 1 study before pursuing clinical testing of CAR T cells. Although DLK1 has minimal expression across most normal tissues, there is high expression in several organs, such as the adrenal gland, particularly the adrenal medulla, compared to the adrenal cortex[10,52]. ADCT-701 treatment may thus lead to adrenal hormone deficiency requiring supplementation with mineralocorticoids and/or corticosteroids. Targeting DLK1 could also have deleterious effects on normal tissue or organ regeneration, as DLK1 is expressed on and regulates many immature stem/progenitor cells, such as hepatoblasts[53]. Another active phase 1 trial targeting DLK1 with an ADC in advanced cancers (NCT06005740)[54] using monomethyl auristatin E (MMAE) as the payload (also an ABCB1 substrate), may also provide additional safety information.

There are several limitations to our study. While we demonstrate that DLK1 is a regulator of ABCB1 expression and thereby sensitivity to a DLK1-directed ADC and chemotherapy, there are likely additional variables which affect ADC and chemotherapy sensitivity that we are unable to account for in this study. Moreover, while we focused on the role of DLK1 in mediating ADC resistance in our functional studies, DLK1 is known to regulate cancer stemness[53,55,56] and tumor progression[27,57]. Thus, further investigation into the role of DLK1 in ACC tumorigenesis will be important. Apart from ACC, we found that DLK1 is more heterogeneously expressed in NE cancers such as SCLC, suggesting further insights into mechanisms of DLK1 expression, particularly epigenetic mechanisms that may control DLK1 imprinting, will be important for the development of potential future DLK1-targeted combination therapies. Nonetheless, as recent parallel work has demonstrated DLK1 as an immunotherapeutic cell surface target in pediatric neuroblastoma[10], a biomarker-based assessment of DLK1 across a broader group of malignancies could be a future clinical approach for DLK1-directed immunotherapeutic clinical trials. Of note, given the challenge of assessing membrane-specific DLK1 expression with current antibodies, alternative detection methods may be required for future DLK1 targeting clinical trials. Lastly, in light of recent studies implicating sexual dimorphism in ACC tumorigenesis[58,59], we cannot rule out the possibility that sex may have a role in ADCT-701 efficacy as our experiments were primarily conducted using female patient-derived models; however, DLK1 expression has not been shown to be correlated with sex or hormonal status in ACC[17].

In summary, we have identified DLK1 as a new immunotherapeutic target in ACC and NE neoplasms such as SCLC. We have also demonstrated that DLK1 is an important driver of chemotherapy and ADC resistance through the regulation of the drug efflux pump ABCB1. Our data support the clinical testing of targeting DLK1 with an ADC in ACC and other NE neoplasms and identify DLK1 as an important cell surface target for future immunotherapeutic approaches.

## Methods

### ACC patient tumor specimens
This work complies with all relevant ethical regulations. All patients provided informed consent, and protocols used for this study were

approved by the Institutional Review Board at the National Institutes of Health (NCT05237934, NCT01109394, and NCT03739827). Tumors were collected from surgical resection of metastatic sites at the NIH Clinical Center and were used for short-term organoid experiments and DLK1 IHC. A summary of ACC patient tumors with associated with DLK1 IHC and/or short-term organoid assay data used in this study are shown in Supplementary Data 1.

### Bulk and single-cell RNA sequencing

For bulk RNA-seq, RNA was extracted from PDX tumors using an All-Prep DNA/RNA extraction kit according to the manufacturer's protocol (Qiagen #80204). Samples were pooled and sequenced on NovaSeq 6000 SP using Illumina total RNA Prep and paired-end sequencing. Samples were trimmed for adapters and low-quality bases using Cutadapt before alignment with the reference genome (hg19) and the annotated transcripts using STAR. Gene expression quantification analysis was performed using STAR/RSEM tools. Read counts for each gene between samples were normalized using the TMM method implemented in edgeR. Differential gene expression was subsequently performed between relapsed and control tumors using DESeq2, following by the generation of volcano plots.

Single cell RNA-seq was performed on single cell suspensions from 18 ACC liver and/or lung metastases were sequenced on the 10x Genomics Chromium Platform targeting 6000 cells per sample. Sequencing was performed on either an Illumina NextSeq 550 or an Illumina NextSeq 2000 instrument. Data was processed using the cellranger pipeline and downstream analysis performed in R using Seurat. Our analysis categorized high and low *NOTCH1* cells (based on median expression) in malignant cells (identified by copy number variation using inferCNV) with non-zero *NOTCH1* expression.

### Tumor cells isolation and short-term organoid culture

To generate short-term organoid cultures, fresh ACC patient tumors were minced into tiny fragments in a sterile dish. Tumor fragments were performed to enzymatic digestion in advanced DMEM/F12 (Gibco, #12491015) supplemented with 1x Glutamax (Gibco, #35050061) and 10 mM HEPES buffer (Quality Biological, #118-089-721) containing collagenase type 4 (200 μl/ml, Worthington Biochemical, #LS004188) and DNase I (50 μl/ml, Worthington Biochemical, #LS006361) on an orbital shaker for 1 hr at 37 °C and filtered through 70 μm strainers. The mixture was spun for 5 min at 1500 rpm. The cell pellet was treated with 1 x RBC lysis buffer (Roche, #11814389001) for 5 min at room temperature to remove the red blood cells and then spun for 5 min at 1500 rpm. Growth Factor Reduced-Matrigel (Corning, #354230) was mixed with tumor cells in minimum basal medium (MBM) consisting of DMEM/F12 (1:1) (1x) (Gibco, #11320033), 1 x N2 supplement (Gibco, #17502048), 1 x B27 supplement (Gibco, #17504044), 50 ng/mL EGF (PeproTech, #100-15R-1MG), 20 ng/mL bFGF (PeproTech, #100-18B-1MG), 100 ng/mL IGF-2 (PeproTech, #100-12-500UG), and 10 μM Y-27632 (STEMCELL Technologies, #72304) at a 1:1 ratio and added to a 6-well plate. Each well was overlaid with 2 ml MBM medium after Matrigel had solidified in a 37 °C and 5% CO₂ culture incubator for 20 min. ACC organoid culture, MBM medium was refreshed once a week. Every 2-4 weeks, organoids were passaged by mechanical pipetting Matrigel gently using Dispase in DMEM/F12 media (STEMCELL Technologies, #07923) and several washes with PBS until Matrigel was cleared out. Organoid fragments were then re-suspended in Matrigel and seeded as described above.

### In vitro short-term organoid culture cytotoxicity assays

Human ACC patient tumor single cells were embedded in 10 μl of MBM medium with 50% Growth Factor Reduced-Matrigel (Corning, #354230) on the 384-well white plate at a concentration of 2000 cells per well. After solidification of the Matrigel for 30 min at 37 °C, 20 μl

fresh MBM medium was added to each well, and the plates were further incubated for 2 days. After the 2 days of pre-culture, cells were treated with 30 μl ADCT-701 and B12-PL1601 for 7 days or with 30 μl SG3199 or other chemotherapeutic drugs for 3 days. For ADCT-701 with or without ABCB1 inhibitors cytotoxicity, NCI-ACC51, NCI-ACC40, and NCI-ACC48 organoid single cells were embedded in Matrigel on 384-well white plates. After 2 days of incubation, cells were treated with different concentration of ADCT-701 or SG3199 (GlpBio, #GC62691) combined with or without 1 μM valspodar (APExBIO, #A3905), 10 μM elacridar (SelleckChem, #S7772), and 1 μM tariquidar (SelleckChem, #S8028) for 7 days or 3 days respectively. To assess the cytotoxicity of ABCB1 inhibitors (1 μM valspodar, 10 μM elacridar, and 1 μM tariquidar) alone, NCI-ACC51, NCI-ACC40, and NCI-ACC48 PDOs were seeded and treated with or without ABCB1 inhibitors for 3 and 7 days. For chemosensitivity of NCI-ACC51, NCI-ACC40, and NCI-ACC48 organoids, cells were plated in 384-well white plates as in the previous seeding steps. After 2 days of incubation, cells were treated with mitotane, etoposide, doxorubicin, or carboplatin for 3 days. 20 μl of CellTiter-Glo reagent (Promega, #G7573) was added, and the luminescence was quantified with a SpectraMax i3x reader (Molecular Devices).

### Cell lines

Human ACC cell lines CU-ACC1 and CU-ACC2 were obtained from the University of Colorado. CU-ACC cells were cultured in 3:1 (v/v) Ham's F-12 Nutrient Mixture (Gibco, #11765054)−DMEM (Gibco, #11965092) containing 10% heat-inactivated FBS (Gemini Bio, #100-106), 0.4 μg/mL hydrocortisone (Sigma-Aldrich, #H0888), 5 μg/mL insulin (Sigma-Aldrich, #I0516), 8.4 ng/mL cholera toxin (Sigma-Aldrich, #C9903), 10 ng/mL epidermal growth factor (STEMCELL Technologies, #78006.1), 24 μg/mL adenine (Sigma-Aldrich, #A8626) and 1% Penicillin-Streptomycin (Gibco, #15140122). Human ACC cell line H295R (CRL-2128) was obtained from ATCC and cultured in 1:1 DMEM:F12 (Gibco, #11320033) containing 2.5% Nu-Serum (Corning, #355100), 1% ITS + Premix Universal Culture Supplement (Corning, #354352), and 1% Penicillin-Streptomycin. Human PCPG cell line hPheo1 was a gift from Dr. Karel Pacak at the Eunice Kennedy Shriver National Institute of Child Health and Human Development (NICHD), NIH and cultured in RPMI-1640 (Corning, #MT10040CM) with 10% FBS and 1% Penicillin-Streptomycin. Human SCLC cell lines H146 (HTB-173) and H524 (CRL-5831) were obtained from ATCC. Human SCLC cell line H1436 was obtained from Haobin Chen (Washington University). H146 and H524 cells were cultured in RPMI-1640 (Corning, #MT10040CM) supplemented with 10% FBS and 1% Penicillin-Streptomycin. H1436 cells were cultured in HITES media DMEM/F12 (1:1) (Gibco, #11320082) containing 5% FBS, 1x Glutamax™ (Gibco, #35050061), 10 nM Hydrocortisone (Sigma-Aldrich, #H6909), 10 nM beta-estradiol (Sigma-Aldrich, #E2758), Insulin-Transferrin-Selenium mix/solution (Invitrogen, #41400045), and 1% Penicillin-Streptomycin. All cell lines were cultured at 37 °C in a humidified incubator with 5% CO₂, regularly tested to be mycoplasma-negative (Lonza, #LT07-318) and authenticated by STR profiling (Laragen Inc.).

### In vitro cell line cytotoxicity assays

ACC or SCLC cells were seeded into a 384-well white plate at a concentration of 1500 cells per well in 30 μl medium and allowed to adhere overnight. The PCPG cell line, hPheo1, 300 cells per well in 30 μl medium, were plated into the 384-well white plate for overnight. 30 μl fresh medium containing different concentrations of antibody-drug conjugate (ADCT-701 and B12-PL-1601) or free payload (SG3199) was added to each well, and the plates were further incubated for 7 days or 3 days, respectively. After 7 days (ADCT-701 and B12-PL1601) or 3 days (SG3199) incubation, 20 μl of CellTiter-Glo reagent was added and the luminescence was recorded using a SpectraMax i3x reader (Molecular Devices).

## RNA extraction and quantitative PCR

Total RNA from human ACC short-term PDOs (NCI-ACC44, NCI-ACC51, NCI-ACC56, NCI-ACC40, NCI-ACC48, and NCI-ACC54) were isolated using a RNeasy® Mini kit (Qiagen, #74104) according to the manufacturer's protocol. RNA quality and quantity were determined by NanoDrop™ One (Thermo Scientific, #ND-ONE-W). 1 μg total RNA was used for cDNA synthesis by High-Capacity cDNA Reverse Transcription Kit (Applied Biosystems, #4374966). The cDNA was used to quantify relative expression of mRNA by real-time PCR using human *GAPDH* (Thermo Fisher, Assay ID: Hs99999905_m1) as a housekeeping control. To evaluate the most common ABC transporter genes, human *ABCB1* (Thermo Fisher, Assay ID: Hs00184500_m1), *ABCC1* (Thermo Fisher, Assay ID: Hs01561483_m1), *ABCG2* (Thermo Fisher, Assay ID: Hs01053790_m1), and *ABCC2* (Thermo Fisher, Assay ID: Hs00960489_m1) probes were used. Real-time PCR was conducted using TaqMan Fast Advanced Master Mix (Applied Biosystems, #4444557) on a QuantStudio™ 5 Real-Time PCR System (Applied Biosystems, #A34322). The expression level was calculated using the subtraction of the GAPDH cycle threshold (Ct) value from the target Ct value (ΔCt).

## IHC staining

4-5 μm sections from formalin-fixed, paraffin-embedded blocks were stained using the Bond Refine polymer staining kit (Leica Biosystems, #DS9800) for DLK1 antibody (dilution 1:2000, Abcam, #ab21682) on the Bond Rx automated staining system (Leica Biosystems) following standard IHC protocols with some modifications. A dilution of 1:2000 was chosen based on testing of DLK1 staining intensity of human placenta (positive control) and neuroblastoma PDX, and normal tissue arrays[10]. Briefly, the slides were deparaffinized and incubated with E1 (Leica Biosystems) retrieval solution for 20 min. The primary antibody was incubated for 1 h at room temperature and no post-primary step was performed. Cover-slipped slides were scanned with an Aperio CS-O slide scanner (Leica Biosystems). DLK1 immunohistochemistry was scored by a pediatric pathologist. Each case was scored for the most prominent intensity (0−3, with 1 representing equivocal, 2 weak, and 3 strong positive staining) as well as for percentage of staining. A modified H-score was calculated as intensity multiplied by percentage of positively stained cells reflecting intensity and distribution.

For Ki67 and synaptophysin immunohistochemistry, auto-stainers Ventana Benchmark Ultra (Ventana, Tucson, AZ) were used. The Leica Bond Max (Leica Biosystems, Deerfield, IL) auto-stainer was used for SF1 immunohistochemistry. Validation of these stains was performed on daily clinical laboratory controls by the Anatomic Pathologist on clinical service at the Laboratory of Pathology, National Cancer Institute. An Agilent Technologies anti-Ki67 antibody (mouse, monoclonal, # M7240, clone MIB-1, Santa Clara, CA) was used at a dilution of 1:200. A Perseus Proteomics anti-SF1 antibody (mouse, monoclonal, #PP-N1665-00, clone N1665, Komaba, Japan) was used at a dilution of 1:200. A Roche Diagnostics anti-synaptophysin antibody (rabbit, monoclonal, # 790-4407, clone SP11, Indianapolis, IN) was used at a prediluted concentration.

## Flow cytometric analysis

For surface DLK1 expression analysis, ACC cells, PCPG cell line hPheo1, ACC PDO cells, ACC PDX cells, and SCLC cells were harvested and washed with FACS buffer (PBS containing 1% BSA and 0.1% sodium azide). Cells were incubated with the anti-human DLK1 primary antibody (AdipoGen Life Sciences, #AG-20A-0070-C100; 1:100 per million cells) at 4 °C for 30 min in the dark. Cells were washed by FACS buffer. Cells were then incubated with PE-conjugated secondary antibody (Invitrogen, #P-852; 1:500) at 4 °C for 30 min in the dark. For surface DLL3 expression analysis, human SCLC cells were collected and washed with FACS buffer. Cells were stained with human DLL3-PE (R&D Systems, #FAB4315P; 10 μl per one million cells) or isotype control antibody (R&D Systems, #IC108P; 10 μl per one million cells) at room temperature for 30 min in the dark. For surface ABCB1 expression analysis, human ACC

cells, ACC patient tumor cells, ACC PDX cells, and human SCLC cells were collected and washed with FACS buffer. Cells were stained with human CD243-PE (BioLegend, #348606; 5 μl per one million cells in 100 μl wash buffer) or isotype control antibody (BioLegend, #400214; 5 μl per one million cells in 100 μl wash buffer) at 4 °C for 30 min in the dark. Cells were washed and then stained with PI (BioLegend, #421301; 1:100) following above antibodies incubation. Living cells were separated as PI-negative cells. To semi-quantitate DLK1 or DLL3 cell surface expression in ACC and SCLC cell lines, cell surface molecules of DLK1 or DLL3 per cell were calculated after subtracting background signal from DLK1 secondary antibody alone (Invitrogen, #P852) or DLL3 isotype control antibody (R&D Systems, #IC108P) respectively by BD Quantibrite Beads PE Fluorescence Quantitation Kit (BD Bioscience, #340495) in accordance with the manufacturer's protocol. Stained cells were acquired on LSR Fortessa (BD Biosciences), and data were analyzed using FlowJo software version 10.8.1. Flow cytometry gating strategies are shown in Supplementary Fig. 11A−C.

## Imaging flow cytometry

A total of $1 \times 10^6$ CU-ACC1 or H295R cells were seeded in each well of a six-well plate and allowed to attach overnight. Then, attached cells were incubated with APC-conjugated DLK1 antibody (R&D Systems, a bio-techne brand, #FAB1144A) or isotype control antibody (R&D Systems, a bio-techne brand, #IC0041A) for 1 h in 2 ml of cell culture media at 37 °C. Then, the cell monolayer was collected and rinsed with cold PBS twice and resuspended. The cellular internalization rate of DLK1 antibody in treated cells was evaluated using an Amnis ImageStreamX Mark II imaging flow cytometry (Luminex, Austin, TX, USA).

## Cell cycle and apoptosis assays

For EdU incorporation studies, cells were processed as per the manufacturer's instructions (Invitrogen, #C10634). Briefly, $3 \times 10^5$ CU-ACC1 or H295R cells were plated in 6-well plates and allowed to adhere overnight. CU-ACC1 or H295R cells were then treated with 0.02 μg/mL ADCT-701 or B12-PL1601 for 2 or 5 days, respectively. Cells were labeled with 10 μM Click-iT™ EdU in a 37 °C and 5% $CO_2$ culture incubator for 1 h. Cells were then fixed and permeabilized. Click-iT™ Plus reaction cocktail was added in cells. Cells were then stained with DAPI for DNA content and detected using an LSR Fortessa cytometer (BD Biosciences) and analyzed using FlowJo software version 10.8.1. For Annexin V staining, $2 \times 10^6$ CU-ACC1 or $1.5 \times 10^6$ H295R cells were seeded in 6 cm dish and treated with 20 μg/mL ADCT-701 or B12-PL1601 for 1 or 3 days respectively. Apoptosis was detected using an FITC Annexin V Apoptosis Detection Kit with PI (BioLegend, #640914) following the manufacturer's instructions. Briefly, cells were washed with cold PBS containing 1% BSA and 0.1% sodium azide and resuspended in Annexin V binding buffer and stained with Annexin V and PI at room temperature and then analyzed immediately. Annexin V-positive cells were detected using an LSR Fortessa cytometer (BD Biosciences) and analyzed using FlowJo software version 10.8.1.

## Immunoblotting

Cells were lysed in RIPA buffer (Millipore, #20-188) supplemented with protease inhibitor (Sigma-Aldrich, #11836153001) and phosphatase inhibitors (Sigma-Aldrich, #04906837001). Protein concentration was determined by the DC™ Protein Assay Reagents Package Kit (Bio-Rad, #5000116). 20 μg of protein lysates were resolved on 4−15% Protein Gel (Bio-Rad, #5671084) and transferred to nitrocellulose membrane. The membranes were blocked in 5% blotting grade blocker (Bio-Rad, #1706404XTU) in TBS with 0.1% Tween-20 and then incubated with the indicated primary antibodies. Primary antibodies (1:1000) included DLK1 (CST, #2069), phospho-Histone H2A.X (Ser139) (Millipore, #05-636), cleaved caspase-3 (Asp175) (CST, #9661), cleaved PARP (Asp214) (CST, #9541), total NOTCH1 (CST, #3608), NOTCH1-ICD (CST, #4147), and SYP (CST, #36406). Primary antibody for detection of α-tubulin (Sigma-Aldrich, #T9026) was used at a dilution of 1:1500. Secondary

antibodies (1:5000) were from donkey anti-rabbit IgG-HRP (Cytiva, #NA934) and sheep anti-mouse IgG-HRP (Cytiva, #NA931).

### In vitro bystander killing assays

Bystander activity was assessed by co-culturing WT CU-ACC1 and CU-ACC1 *DLK1* KO (clone 10) cells at various ratios in white-walled 384-well tissue culture-treated plates with complete media. The following day, cells were treated with 1 μg/mL ADCT-701 or B12-PL1601 and incubated in a humidified atmosphere with 5% $CO_2$ at 37 °C for 4 days. Cell viability was measured by the CellTiter-Glo Luminescent Cell Viability Assay kit (Promega, #G7573). Bystander activity was also assessed using conditioned media assays in which CU-ACC1 cells were seeded at a density of 1500 cells/well. ADCT-701 was then added in triplicate the next day, in a dose titration ranging from 20 μg/mL to 20 pg/mL. Cells were subsequently incubated for 5 days. On day 5, 30 μL of conditioned media from these plates was removed from each well and transferred to a fresh plate containing CU-ACC1 *DLK1* KO (clone 10) or WT CU-ACC1 cells, which were plated 24 h previously in 30 μL complete media (final volume 60 μL). These plates were incubated for 5 days before cell viability measurement. Cell viability was determined using the CellTiter-Glo Luminescent Cell Viability Assay kit, and data were presented as percent cell viability relative to untreated controls.

### Lentiviral constructs and lentivirus production

For the CRISPR-Cas9 system, a single target sequences for CRISPR interference were designed using the sgRNA designer (https://portals.broadinstitute.org/gppx/crispick/public) and subcloned into the lentiCas9-Blast (Addgene, #52962). Viral transduction was performed in the presence of polybrene (5–10 μg/mL, Sigma-Aldrich, #TR-1003-G), and cells were centrifuged at 1200 x g for 4 h at 30 °C followed by removal of virus and polybrene. After 72 h, cells were selected with blasticidin (1–4 μg/mL) for 5 days.

### siRNA-mediated knockdown of DLK1

DLK1-targeting siRNA (siDLK1-1: #4392420 ID s16740, siDLK1-2: #4392420 ID s16738, siDLK1-3: #4392420 ID s16739) and control siRNA (#4390843) were purchased from Invitrogen. CU-ACC1 ($5 \times 10^5$ cells/well) cells were plated in a 6-well plate overnight. Cells were then transfected with siRNA at a final concentration of 30 nM using Lipofectamine 3000 (Invitrogen, #L3000015). Three days after transfection, whole cell lysates were collected and analyzed by Western blotting.

### Mice

All animal procedures reported in this study were approved by the NCI Animal Care and Use Committee (ACUC) and in accordance with federal regulatory requirements and standards. All components of the intramural NIH ACU program are accredited by AAALAC International. Six-eight week old female NOD-*scid* IL2Rg$^{null}$ (NSG) mice were obtained from the CCR Animal Research Program for SCLC and ACC xenograft and ACC PDX ADCT-701 in vivo experiments. Both male and female 6-8 week old NSG mice were used for ACC PDX passaging. Seven-week-old female (Crl:NU(NCr)-*Foxn1*$^{nu}$) athymic nude mice were purchased from Jackson Labs. Mice were housed in IVM Microisolator conditions designed for immune compromised mice at 22 degrees Celsius with humidity of 50% and a 12 h light-dark cycle (6 am - 6 pm).

### In vivo efficacy studies

For ACC cell line-derived xenograft models, $2 \times 10^6$ CU-ACC1 or H295R cells were subcutaneously injected into the right flank of female 6–8 week old NSG mice. When the tumor volume reached approximately 100–150 mm³, mice were randomized to each group (3-4 mice per group). For ACC PDX models, 164165 and 592788 were obtained from the NCI Patient-Derived Models Repository (PDMR) within the NCI Developmental Therapeutics Program. POBNCI_ACC004 PDX was developed by the NCI Pediatric Oncology Branch. 164165 and 592788

PDX tumor fragments were implanted subcutaneously into the right flank of female 6-8 week old NSG mice by using a trocar needle. $2 \times 10^6$ POBNCI_ACC004 PDX single cells were injected subcutaneously into female 6–8 week old NSG mice. Recruitment of paired mice in equal numbers to treatment groups was staggered as necessary for any given study. Mice were randomized to each group (5–8 mice per group) once tumors reached 100–200 mm³. For SCLC cell line-derived xenograft models, $2 \times 10^6$ H524 or H1436 cells were implanted subcutaneously into the right flank of male or female 6–8 week old NSG mice. $2 \times 10^6$ H146 cells were injected subcutaneously into right flank of female 7 week old athymic nude mice. Tumor-bearing mice were randomized into three treatment groups (4-7 mice per group) once tumor volume reached 100–150 mm³. All tumor cells used in vivo were suspended in 100 μL of PBS with 50% Matrigel (BD #356237). Normal saline, B12-PL-1601 (1 mg/kg, diluted in normal saline), and ADCT-701 (1 mg/kg, diluted in normal saline) were intravenously injected into the tail vein on day 0 based on a previously established in vivo dosing protocol for ADCT-701[18]. PDX tumor-bearing mice that initially responded to ADCT-701 treatment were re-treated with ADCT-701 if the re-growth tumor volume reached over 100 mm³ (otherwise no further doses were administered). Xenograft tumor-bearing mice that did not initially respond to ADCT-701 were re-treated at D7. Body weight and tumor size were measured once or twice weekly, respectively, and tumor volume (mm³) were calculated with the formula as length x width²x 0.5. The maximum tumor volume allowed through our IACUC is 2000 mm³. Mice were euthanized when tumor volume reached 1500–2000 mm³ or 100 days after dosing. Tumors were collected for RNA-sequencing and IHC analysis.

### Quantification and statistical analysis

All statistical tests between groups were unpaired two-tailed Student's *t* tests, unless otherwise stated, and *p*-values less than 0.05 were considered statistically significant. For box plots, the horizontal line represents the median, the lower and upper boundaries correspond to the first and third quartiles, and the lines extend up to 1.5 above or below the IQR (where IQR is the interquartile range, or distance between the first and third quartiles). GraphPad Prism 10.3.1 was used for all graphing and statistical analyses.

### Reporting summary

Further information on research design is available in the Nature Portfolio Reporting Summary linked to this article.

## Data availability

The bulk RNA-seq data generated in this study have been deposited in the Gene Expression Omnibus (GEO) database under accession code GSE294644. The previously published expression datasets re-analyzed in this study can be accessed in the GEO database under accession code GSE10927 and in the Genomics Data Commons Data Portal at https://portal.gdc.cancer.gov/. Single-cell RNA-seq data has been deposited in the database of Genotypes and Phenotypes (dbGaP) under accession code phs003987 (https://www.ncbi.nlm.nih.gov/projects/gap/cgi-bin/study.cgi?study_id=phs003987). The remaining data are available within the Article, Supplementary Information or Source Data file. Source data are provided in this paper.

## Code availability

No custom code or mathematical algorithms were used in this work.

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

## Acknowledgements

We thank the technicians in the CCR Animal Research Program for their support of this study. This work is funded by the NIH Intramural Research Program (ZIA BC011989 N.R.), ADC Therapeutics (N.R. and J.R.), Department of Defense Rare Cancers Research Program Concept Award (S.K. and N.R.), the My Pediatric and Adult Rare Tumor Network (MyPART) (K.R. and B.W.), and R35 CA220500 (J.M.M.).

## Author contributions

Study conception and design: NY.S. and N.R.; Tissue and clinical data collection: NY.S., S.K., Y.S.K., D.V., A.M., R.N., R.O., K.R., B.W., D.T., J.D.R., and N.R. In vitro and in vivo experiments: NY.S., S.K., and YS.K.; Analysis and interpretation of experimental data: NY.S. and N.R.; Immunohistochemistry staining and analysis: M.P., D.M., A.K.H., S.J.D., J.M.M., and J.P.; Bulk RNA-seq analysis: F.E., A.D., Y.P., and N.R.; Single cell RNA-seq analysis: E.A., C.CB., R.K., and N.R.; Manuscript writing: NY.S. and N.R.; Manuscript revisions: NY.S., K.KV., M.E.W, R.W.R, M.M.G., and N.R. All authors reviewed the results and approved the final version of the manuscript.

## Funding

## Competing interests

Nitin Roper and Jaydira Del Rivero have received research funding from ADC Therapeutics for this study. The other authors have no competing interests to report.

## Additional information

[1]Developmental Therapeutics Branch, Center for Cancer Research, NCI, Bethesda, MD, USA. [2]Pediatric Oncology Branch, Center for Cancer Research, NCI, Bethesda, MD, USA. [3]Laboratory of Pathology, Center for Cancer Research, NCI, Bethesda, MD, USA. [4]Department of Medicine-Endocrinology/Metabolism/Diabetes, University of Colorado, Anschutz Medical Campus, Aurora, CO, USA. [5]Research Service Rocky Mountain, Regional Veterans Affairs Medical Center, Aurora, CO, USA. [6]Department of Pathology and Laboratory Medicine, The Children's Hospital of Philadelphia and Perelman School of Medicine, University of Pennsylvania, Philadelphia, PA, USA. [7]Department of Pediatrics, Perelman School of Medicine, University of Pennsylvania, and Children's Hospital of Philadelphia, Philadelphia, PA, USA. [8]Laboratory of Cell Biology, Center for Cancer Research, NCI, Bethesda, MD, USA. ✉e-mail: nitin.roper@nih.gov

