## [Transparent Peer Review file · Nature Communications]

Identification of the Notch ligand DLK1 as an immunotherapeutic target and regulator of tumor cell plasticity and chemoresistance in adrenocortical carcinoma

Corresponding Author: Dr Nitin Roper

Version 0:

Reviewer comments:

Reviewer #1

(Remarks to the Author)

In this study, the authors uncover delta-like non canonical Notch ligand 1 (DLK1) as a cell surface protein with high expression in refractory adult metastatic cancers including small cell lung cancer (SCLC) and adrenocortical carcinoma (ACC), identify DLK1 as a novel immunotherapeutic target that regulates tumour cell plasticity and chemoresistance in ACC, and support targeting DLK1 with an antibody-drug conjugate (ADC) in ACC and neuroendocrine neoplasms in an active first-in-human phase I clinical trial (NCT06041516).

The manuscript is well-written, and the results are certainly of great interest in the field of rare aggressive cancers with extremely limited therapeutic options such as ACC.

However, we have some significant concerns regarding the characteristics of the selected adrenal tumour samples (and relative clinical phenotype), conclusiveness of experiments used to support a functional link between DLK1, ABCB1 and drug resistance, and a lack of background literature and references that could lead readers to misinterpret the novelty of many claims put forward by the current paper.

Detailed major and minor comments are reported below.

Major comments:

1) DLK1 protein expression by IHC: the authors provide DLK1 IHC analysis in n=29 ACC metastatic tumours and report positive expression in 97% of cases (widely ranging from 10 to 300 H score). Hereby, a better description of the staining distribution (i.e. nuclear vs cytoplasmatic, homogeneous vs heterogeneous, etc) would be recommended. We also notice that the scoring was done manually by one single operator, while it is good practice to have two independent blind scoring operators and an agreement pathway in place. In addition, a comment regarding the specificity of the DLK1 antibody (dilution 1:2000) would be useful.

Moreover, a recent paper available in BioRxiv 2024 (Mariniello et al, <https://doi.org/10.1101/2024.08.22.609117>) shows the results of IHC expression in a larger and more diverse cohort of 73 + 159 ACC patients showing moderate expression in localized ACC and higher expression in metastatic disease. This should be extensively discussed (see also our point 3 below).

2) Role of ABCB1 in drug tolerance/resistance and its regulation by DLK1: Whilst evidence for specific targeting of ADCT-701 to cells expressing DLK1 is clear, perhaps the weakest part of the current paper is experimental support for the claim that DLK1 is a "major" regulator of ABCB1. Moreover, that this somehow confers the delicate balance required for an ideal response to treatment with ADCT-701, called "counterintuitive" by the authors in the discussion. Starting on line 202, some concerns that likely require some further experiments to address include: (i) why just two specific responder PDOs (NCI-ACC51 and NCI-ACC40) were selected in Figure 3A, when in principle all six could be tested with PBD; (ii) interpreting the difference in gene expression for ABCB1 and ABCG2 as significant (relative to the other ABC genes) based on a two-group comparison with just N=1 sample in either group (NCI-ACC48 and NCI-ACC51). Again, here differential gene expression analysis between DLK1+ responder and non-responder groups could be done correctly if all PDOs are included; (iii) lack of controls for toxic effects of ABCB1 inhibitors (valspodar, elacridar or tariquidar) by exposure of PDOs to these compounds in the absence of SG3199 or ADCT-701 in Figures 3D and S7B. Without these controls, toxicity of the inhibitors alone could explain the observed decrease in cell viability, rather than induction of SG3199/ ADCT-701 sensitivity through ABCB1 inhibition as is being claimed. The same concern about the lack of such a control applies to identical experiments performed

with PDX models, as presented in Figures 3F, 3G and 3J.

The lack of ABCB1 surface expression in DLK1 KO cells reported in Figure 5E is striking but raises some concerns about the biological relevance versus artefactual interpretation of this observation, which should be tested further. As the authors discuss and demonstrate, expression of CYP17A1 is also downregulated in DLK1 KO cells, which might be expected based on the known associations between DLK1 and adrenocortical differentiation (see point 3 below). It follows that the correlation between expression levels of DLK1 and CYP17A1 should therefore be strongly positive in normal adrenal tissue (for example, in the GTEx adrenal dataset). However, by the same logic, one would similarly expect to find a strong positive correlation between DLK1 and ABCB1 expression levels if there is a previously undescribed, coregulation of these two genes also. We therefore highly recommend that the authors perform such an analysis, comparing the correlation between DLK1, CYP17A1 and ABCB1 in adrenal GTEx data (as they did with DLK1 and NOTCH1 in ACC), to establish whether the regulatory association proposed based on cell line experiments is biologically relevant by this criterion. If this turns out not to be the case, a stronger argument will need to be made to justify why decreased ABCB1 expression is not a technical artefact arising during DLK1 KO cell line preparation.

3) DLK1 in adrenocortical cell differentiation, stem/progenitor cells and ACC: The authors discuss the role of DLK1 as driver of adrenocortical cell differentiation and the relevance of this for the biology of ACC. However, they describe this out of the context of and without reference to a number of previously published results on the same topic. In these studies, the role of DLK1 in the homeostatic maintenance of the gland has already been investigated extensively. It has been proposed that the human adrenal cortex remodels with age to generate clusters of relatively undifferentiated cells expressing DLK1 (named DLK1-expressing cell clusters or DCCs), potentially representing a novel cell population in the human adrenal cortex.

Therefore, we recommend discussing the interpretation of any new data on this topic in the context of the following publications:

Guasti L, et al. Dlk1 up-regulates Gli1 expression in male rat adrenal capsule cells through the activation of β 1 integrin and ERK1/2. *Endocrinology*. 2013 doi: 10.1210/en.2013-1211.

Hadjidemetriou et al. DLK1/PREF1 marks a novel cell population in the human adrenal cortex. *J Steroid Biochem Mol Biol*. doi: 10.1016/j.jsbmb.2019.105422.

Pittaway et al. The role of delta-like non-canonical Notch ligand 1 (DLK1) in cancer. *Endocr Relat Cancer*. 2021 doi: 10.1530/ERC-21-0208.

Mariniello et al. Dlk1 is a novel adrenocortical stem/progenitor cell marker that predicts malignancy in adrenocortical carcinoma. *bioRxiv [Preprint]*. 2024 Aug 22:2024.08.22.609117. doi: 10.1101/2024.08.22.609117.

One of these (Hadjidemetriou et al) is cited on line 410 of the current manuscript but only in the context of levels of DLK1 expression in the adrenal cortex. We feel this does not do justice to any of the above listed studies that all put forward mechanistic roles for DLK1 in adrenal/ACC. In absence of appropriate referencing, this may give the reader a false impression any such claims are entirely novel to the current paper.

4) ACC patient tumour specimens and clinical data: We appreciate that the focus of the study is on advanced (metastatic) ACC. However, the authors should explain the rationale behind their choice to not include any primary tumours, at least for a comparison with metastatic lesions.

Moreover, the authors only show the demographic and clinical data relative to the cohort used for short-term organoid experiments (n=13, Suppl Table 3). We recommend also displaying data for the entire cohort, i.e. including cases used for RNA-seq (n=46) and IHC (n=24). In fact, this information would be relevant for interpreting the biological relevance of the gene/protein expression data and response to treatment (e.g. evaluating the relationship between pre-treatment with mitotane or systemic drugs, and the steroid secretion profile, especially cortisol, or aggressiveness of the tumour etc.). Additionally, the authors should also include among the presented data also the site of metastatic tumours tested in each experiment as well as the time passed from the primary surgery to the first disease recurrence and the surgery for metastatic disease.

Finally, regarding Suppl Table 3, there are few open issues to be improved/clarified, including: do the ki67% and the hormonal status data correspond to the primary tumour or metastasis? How many and which samples for each patient have been tested (the ID numbers do not always correlate with the colour code)? We strongly suggest revising this table to include such information.

5) Bulk and single-cell RNA sequencing: In the methods section starting on line 461, the authors indicate that all bulk RNA-seq data generated in the current study involved exclusively genetic material isolated from FFPE. Could they comment on the RNA-seq quality control for readers to assess the reliability of these results? To our knowledge, generating RNA-seq data from FFPE can be notoriously difficult and results must be carefully validated during quality control. Somewhat confusingly, immediately after this statement, the authors also say: "RNA-seq libraries were prepared using Illumina TruSeq RNA Access Library Prep Kit or Total RNA Library Prep Kit according to the manufacturer's protocol (Illumina)". Which one of these two kits was used and, if a combination of both, why was this so and what are the batch effects of using one kit compared to the other? Illumina's own documentation highlights that not all their kits (e.g. TruSeq RNA Library Prep Kit v2) are compatible with FFPE extractions.

On several occasions (e.g. lines 323 and 470) the authors refer to the study Aber et al. manuscript in submission when, to our knowledge, no manuscript is available in either published or preprint form at the time of the current review. This is problematic, since the Aber et al study is used both to justify some major claims made in the current paper in addition to detailing the protocol for single-cell RNA-seq with metastatic ACC samples. Specifically, a claim is made by the authors that these single-cell data support the hypothesis that DLK1 expression is higher in populations of cells that are also enriched for the adrenal differentiation score (ADS), which is interpreted as a causal role for DLK1 in adrenocortical differentiation (this is emphasised on line 53-54 of the abstract, which indirectly references Aber et al, and is also related to our point 3 above).

Beyond the lack of any publicly-available accompanying article by Aber et al to cross-refer to, we see two additional major problems with this:

- i) DLK1 itself is a member of the ADS gene set and so, by definition, there is covariation between its expression levels and the other genes within the set—simply removing DLK1 from the calculation of ADS presented in Fig 5K will not separate this (predefined) correlation from causation and such statistical comparisons to report p values appears unfounded (the ADS scores for DLK1 high and low groups are not necessarily independent);
- ii) at the current time, we are aware of at least two other studies in published/preprint form that provide single-cell resolution data on ACC tumours but are not discussed/cited by the current study:

Tourigny et al Cellular landscape of adrenocortical carcinoma at single-nuclei resolution. *Mol Cell Endocrinol*. 2024 doi: 10.1016/j.mce.2024.112272

Popova et al Single nuclei sequencing reveals replication stress in adrenocortical carcinoma bioRxiv [Preprint]. 2024. doi: 10.1101/2024.09.30.615695

The first of these (Tourigny et al) investigates the role of the imprinted DLK1/MEG3 locus in ACC and the second (Popova et al) focusses specifically on advanced and metastatic ACC samples, as does Aber et al and the current study. At the very least, it would be highly appropriate to discuss the single cell results in the context of and with reference to these previous studies.

6) Male/Female ratio: The authors utilised only female mice (n=7) and a large majority of female patients for the short-term organoid experiments (n=12 out of 13). Even if there is a slight preponderance of females among ACC patients, it is not observed at this level (more than 90%). Moreover, it is known that pathogenic mechanisms involved in tumour development and progression might be different between males and females (i.e. through androgen-dependent induction of senescence, recruitment, and differentiation of highly phagocytic macrophages), see:

Wilmouth et al Sexually dimorphic activation of innate antitumor immunity prevents adrenocortical carcinoma development. *Sci Adv*. 2022 doi: 10.1126/sciadv.add0422.

Warde et al. Senescence-induced immune remodeling facilitates metastatic adrenal cancer in a sex-dimorphic manner. *Nat Aging*. 2023 Jul doi: 10.1038/s43587-023-00420-2.

We would like to ask the authors to consider whether relevant female preponderance could have affected the Notch/DLK1 expression data and/or the ADC in vitro and in vivo efficacy. The authors should justify their choice of focusing on females and discuss potential limitations. This is particularly relevant since DLK1 is an imprinted gene (an observation surprisingly not mentioned or discussed anywhere in the current study) and there are some existing theories that imprinting of maternal/paternal alleles may be directly related to selection for sex-specific function. See, for example:

Patten et al The evolution of genomic imprinting: theories, predictions and empirical tests. *Heredity* 2014 doi: 10.1038/hdy.2014.29

Minor comments:

- Introduction: add a reference for the sentence “recent FDA approvals across a diverse set of malignancies”
- Adrenocortical Differentiation Score: We appreciate that ADS that is taken from the TCGA paper (Zheng et al 2016). However, a better description of the meaning and role of this score in this context would be useful
- Discussion (page 14 line 364): a word after “driver of adrenocortical ...” is missing
- One line 83, the authors say: “to our knowledge, there has been no systematic effort to assess whether Notch ligands beyond DLL3 may or may not be targetable cell surface proteins in cancer”. This sounds contradictory, since they cite reference 23 (preprint by Weiner et al now published as Hamilton et al *Cancer Cell* 2024 doi: 10.1016/j.ccell.2024.10.003), which has identified DLK1 as a surface target in neuroblastoma and has overlapping coauthors with the current study
- We recommend updating the citation from above point to published version in *Cancer Cell* (author names have changed, which can add confusion)
- At the end of the second results section, further cell death is proposed to occur via “bystander killing” mediated by ADCT-701 treatment. Could the authors expand upon what this phenomenon is more generally, and speculate about possible mechanisms?
- On lines 179-181, it is not immediately intuitive how PDOs are classed as DLK1+ or DLK1- based on Figures 2G and H. Perhaps the authors could briefly clarify in that same statement the features of plots that allow this classification
- In Figure 2I (and potentially 2J as well) the two plots appear to be different sizes even though the y axes should be the same. This should be corrected. It also makes direct comparison of the two cell lines difficult
- Several references cited on page 14 seem to be mixed up and not corresponding to adequate papers (e.g. 30-36), which makes the reading difficult to fully understand. Please check
- The word “differentiation” is missing after “adrenocortical” on line 364

Reviewer #2

(Remarks to the Author)

Dear editor,

In this work by Yun Sun et al entitled "Identification of DLK1, a Notch ligand, as an immunotherapeutic target 1 and regulator of tumor cell plasticity and chemoresistance in adrenocortical carcinoma" The authors describe the therapeutic activity of a novel humanized ADC against DLK1 in ACC and SCLC preclinical models and provide evidence that DLK1 notch ligand may represent a promising target for ADC therapy in these two particular indications, both of which represent hard to treat conditions.

Moreover they undiscover the role of the transporter ABCB1 as major player in resistance mechanism to the ADC.

Intriguingly, it's proposed that the therapeutic target DLK1, is directly involved in the negative regulation of Notch signaling and positive regulation of ABCB1 expression which in turn increase chemoresistance. Therefore authors propose that this novel ADC-based therapeutic strategy may be restricted to DLK1 + tumors but with low/medium expression. Importantly, this novel ADC against DLK1 already entered in trial validation.

Overall, this is a solid work based on a consistent bulk of data. Indeed, generally experiments are well designed and conducted.

I would suggest publishing this manuscript once some aspects I raise here have been clarified:

-A complete analytical characterization of this novel humanized ADC should be included in the manuscript

-The rationale behind the choice of dose and schedule relative to ADC treatment should be discussed.

-Figure 2C/4E: ADCT701 in vitro cell killing activity is shown. IC50 calculation is missing.

-Figure 3A-C. Authors clearly demonstrated that there's an upregulation of ABCB1 transporter gene in ACC patients. In parallel, this upregulation has been confirmed in insensitive but not sensitive cell lines thus suggesting that ABCB1 upregulation may be involved in SG3199 resistance. However, authors should explain why ACC48 cells has been selected as principal model for all the further experiments and not ACC40 cell line model given the fact that the former one resulted as the most insensitive to the ADC. Moreover, ACC40 cells do express very high amount of DLK1 (Figure 2H). Is, as expected, ABCB1 upregulated in these cells?

-Figure 3J: statistic is missing. Indeed, re-sensitization to PBD by ABCB1 inhibitors seems not to be so relevant here.

Authors are invited to comment on this.

-Figure 5. Immunoblot and/or mRNA data showing DLK1 knock-out in the selected clones are missing.

To validate DLK1KO data authors used DLK+ ACC48 and DLK- ACC49 cell models. Again, ACC40 cell model would be useful here, as these cells express very high amount of DLK1. Including a second DLK1 high expressing model would reinforce author's findings.

In view of the potential relevance of the role of DLK1 in inducing ADC and chemoresistance, it would be of great interest to evaluate this phenomenon in other than rare ACC tumors. In particular, as example it is known that in ovarian cancer overexpression of DLK1 is associated with a worse prognosis. It would be important to verify whether regulation of ABCB1 by DLK1 is also found in this type of tumor. This information will define whether the mechanisms described here are limited to ACC tumors or shared by other DLK+ malignancies.

Reviewer #3

(Remarks to the Author)

In this paper, the authors identify DLK1 as a potential immunotherapeutic target in ACC and describe some insights into the role of this gene in tumor biology and chemoresistance. Although similar data, using the same ADC and similar methodology, have recently been published in neuroblastoma (Cancer Cell, <https://doi.org/10.1016/j.ccell.2024.10.003>), which I suggest should be added to the manuscript discussion and referenced, the manuscript presents for the first time consistent data in ACC cell lines. The manuscript is clearly written, uses appropriate methodology to support the conclusions, and provides interesting information to the readers of Nature Communication.

I have a few points to address:

1) The DLK1 gene is paternally expressed and related to the Dlk1-Dio3 imprinted domain and regulated by a complex epigenetic process controlled by an imprinted control region (IG-DMR) that is hypomethylated on the maternal allele and hypermethylated on the paternal allele (doi.org/10.1093/nar/gkac344). I think it is important to describe better this genetic particularity and how it may interfere with tumor biology and potential treatment.

2) Although DLK1 is expressed in many human tissues during embryonic development, in adults its expression is mainly restricted to (neuro)endocrine tissues and other immature stem/progenitor cells, being important in several physiological mechanisms. In my opinion, the potential deleterious effects of an ADC against this gene could be discussed.

3) The evidence for the nature of the effect of DLK1 on Notch signaling is conflicting, with some results opposing those described in this manuscript. In high-grade serous carcinoma of the ovary (<https://doi.org/10.1038/s41388-018-0658-5>) and in NSCLC cell models (10.3892/ol.2017.6019), unlike what was found by the authors, a positive relationship between DLK1 and NOTCH1 was described. I think it would be interesting to describe in the SCLC model used the relationship between DLK1 and NOTCH1 and further discuss the conflict association between these genes.

4) The hPheo1 cell line described in lines 139-141 and supplementary figure 2 is not described in the methods

5) Lines 170-171. The authors described that "Overall, these results indicate that ADCT-701 not only can target DLK1+ cells but can also indirectly induce cytotoxicity in DLK1- cells". This potential indirect mechanism can be further discussed. The same goes for lines 180-181: "However, among ADCT-701 non-responders, 50% (n=3/6) were still DLK1+ (Fig. 2H) suggesting that ADCT-701 sensitivity is influenced by factors other than DLK1 expression". Which factors? 6) It is not clear

from the text whether all in vitro experiments were performed in at least three independent experiments, in triplicate. In some of them, the SD bars in the figures are not visible: Fig. 2 C and E (for NCI-ACC-47); Fig. 2 F (for NCI-ACC-48); Fig. 3 D and F; Fig. 4 D and E (for H524); Fig. 5 D; Supplementary Fig. 2 A and D; Supplementary Fig. 3 E, Supplementary Fig. 7 E, Supplementary Fig. 10 I and J (for VP16 and doxo) and K. Please clarify.

7) There are some errors and inconsistencies in the figures. Figure 1E is not described in the figure legend. Figure 3 G, please standardize the formatting and drug description. Supplementary Fig. 10, the letters did not match in the figure, e.g., legend "(D) Photomicrographs of CU-ACC1 parental and DLK KO clones 9 and 10 cells", in Figure D is an anti-ABCB1 histogram. Figure K is not described in the legend. Please review.

Reviewer #4

(Remarks to the Author)

Version 1:

Reviewer comments:

Reviewer #1

(Remarks to the Author)

The Authors have extensively revised the manuscript according to reviewer's requests and satisfactorily replied to the large majority of my comments. I appreciate that the current version is significantly improved with clearer conclusions.

I only have a minor request, i.e. to add to add a comment regarding the DLK1 antibody as short sentence within the limitations of the study.

Reviewer #2

(Remarks to the Author)

Dear Editor,

In my opinion this manuscript can be published in the current form. Authors have successfully addressed all the criticisms raised.

Reviewer #3

(Remarks to the Author)

I consider all questions and suggestions made by this reviewer to have been adequately answered in the revised form, with no new comments or questions.

Reviewer #4

(Remarks to the Author)

REVIEWER COMMENTS

Reviewer #1 (Remarks to the Author): with expertise in adrenocortical tumors

In this study, the authors uncover delta-like non canonical Notch ligand 1 (DLK1) as a cell surface protein with high expression in refractory adult metastatic cancers including small cell lung cancer (SCLC) and adrenocortical carcinoma (ACC), identify DLK1 as a novel immunotherapeutic target that regulates tumour cell plasticity and chemoresistance in ACC, and support targeting DLK1 with an antibody-drug conjugate (ADC) in ACC and neuroendocrine neoplasms in an active first-in-human phase I clinical trial (NCT06041516).

The manuscript is well-written, and the results are certainly of great interest in the field of rare aggressive cancers with extremely limited therapeutic options such as ACC. However, we have some significant concerns regarding the characteristics of the selected adrenal tumour samples (and relative clinical phenotype), conclusiveness of experiments used to support a functional link between DLK1, ABCB1 and drug resistance, and a lack of background literature and references that could lead readers to misinterpret the novelty of many claims put forward by the current paper.

We appreciate the positive comments from the Reviewer on the quality and importance of our manuscript. We agree with the concerns raised and have addressed each comment (both major and minor) below in blue.

Detailed major and minor comments are reported below.

Major comments:

1) DLK1 protein expression by IHC: the authors provide DLK1 IHC analysis in n=29 ACC metastatic tumours and report positive expression in 97% of cases (widely ranging from 10 to 300 H score). Hereby, a better description of the staining distribution (i.e. nuclear vs cytoplasmatic, homogeneous vs heterogeneous, etc) would be recommended. We also notice that the scoring was done manually by one single operator, while it is good practice to have two independent blind scoring operators and an agreement pathway in place. In addition, a comment regarding the specificity of the DLK1 antibody (dilution 1:2000) would be useful.

We have now clarified in the first section of the Results and in the IHC section of the Methods that the DLK1 IHC staining distribution is both cytoplasmic/membranous. We have used the commonly applied estimated H-score (combination of intensity and distribution) because it is difficult to demonstrate both intensity and diffuseness of DLK1 staining as a single numerical value to cover a large number of samples. Our main

conclusion is that most ACC show some degree of DLK1 expression that spans a wide score range, which is supported by other methods within this manuscript (such as DLK1 flow cytometry) as well as from DLK1 IHC in the Marinello et al. 2025 paper. The specific numerical results of the DLK1 H-scores are not correlated with any other findings in our manuscript. We agree that reproducibility is important if an antibody is being used as a diagnostic tool by pathologists or if it is designed as a companion diagnostic for treatment. However, in this work we are not using DLK1 IHC staining in those ways. Moreover, the DLK1 antibody is polyclonal and specific H-score numbers are not likely to be fully reproducible when repeated with another lot of this antibody.

We have also added a comment regarding DLK1 antibody dilution to the Methods section: “A dilution of 1:2000 was chosen based on testing of DLK1 staining intensity of human placenta (positive control) and neuroblastoma PDX and normal tissue arrays (Hamilton et al. *Cancer Cell* 2024).”

Moreover, a recent paper available in BioXRiv 2024 (Mariniello et al, <https://doi.org/10.1101/2024.08.22.609117>) shows the results of IHC expression in a larger and more diverse cohort of 73 + 159 ACC patients showing moderate expression in localized ACC and higher expression in metastatic disease. This should be extensively discussed (see also our point 3 below).

We have now added a sentence in our Results section to reference the DLK1 IHC data from this parallel work. We also discuss this manuscript further as noted below in point 3.

“To validate *DLK1* expression, we performed DLK1 IHC across our cohort of ACC metastatic tumors and found 97% (n=28/29) of ACC patients were DLK1⁺ for cytoplasmic/membranous scoring (mean H-score 147, range 10-300) reflecting a mix of intensity and distribution of DLK1 in ACC (Fig. 1D), which is similar to recent DLK1 IHC data from a large cohort of ACC patients (Marinello et al. 2025).”

2) Role of ABCB1 in drug tolerance/resistance and its regulation by DLK1: Whilst evidence for specific targeting of ADCT-701 to cells expressing DLK1 is clear, perhaps the weakest part of the current paper is experimental support for the claim that DLK1 is a “major” regulator of ABCB1. Moreover, that this somehow confers the delicate balance required for an ideal response to treatment with ADCT-701, called “counterintuitive” by the authors in the discussion. Starting on line 202, some concerns that likely require some further experiments to address include:

(i) why just two specific responder PDOs (NCI-ACC51 and NCI-ACC40) were selected in Figure 3A, when in principle all six could be tested with PBD;

(ii) interpreting the difference in gene expression for ABCB1 and ABCG2 as significant (relative to the other ABC genes) based on a two-group comparison with just N=1 sample in either group (NCI-ACC48 and NCI-ACC51). Again, here differential gene expression analysis between DLK1+ responder and non-responder groups could be done correctly if all PDOs are included;

We fully agree with the Reviewer that the data in Figure 3A and Figure 3B (now Figure 4A and 4B) would be stronger with the use of additional models. Thus, we have now included data using a total of 6 PDO models: 3 DLK1+ responders (NCI-ACC44, NCI-ACC51 and NCI-ACC56) and 3 DLK1+ non-responders (NCI-ACC40, NCI-ACC48, and NCI-ACC54). We chose these 6 PDO models because they were able to grow for multiple passages and allowing us to perform PBD cytotoxicity as well as extract RNA for ABC gene transporter gene expression. Unfortunately, we were unable to continue growing NCI-ACC17 and NCI-ACC52 and therefore could not perform all of the above experiments with these models. Nonetheless, these new data demonstrate that *ABCB1* gene expression is significantly higher in DLK1+ responders compared to DLK1+ non-responders (Figure 4B). As validation of these data, we also show surface ABCB1 expression to be significantly higher in DLK1+ non-responders compared to DLK1+ responders (Figure 4C).

Figure 4

(iii) lack of controls for toxic effects of ABCB1 inhibitors (valsopodar, elacridar or tariquidar) by exposure of PDOs to these compounds in the absence of SG3199 or ADCT-701 in Figures 3D and S7B. Without these controls, toxicity of the inhibitors alone could explain the observed decrease in cell viability, rather than induction of SG3199/ADCT-701 sensitivity through ABCB1 inhibition as is being claimed. The same concern about the lack of such a control applies to identical experiments performed with PDX models, as presented in Figures 3F, 3G and 3J.

To address this concern regarding potential toxicity of ABCB1 inhibitors, we have performed cell viability experiments across our models with ABCB1 inhibitors alone. As shown in Supplementary Fig. 7D and Supplementary Fig. 8D, ABCB1 inhibitors do not

have significant toxicity in 6 different models (PDOs and PDXOs) at two timepoints (D3 and D7).

The lack of ABCB1 surface expression in DLK1 KO cells reported in Figure 5E is striking but raises some concerns about the biological relevance versus artefactual interpretation of this observation, which should be tested further. As the authors discuss and demonstrate, expression of CYP17A1 is also downregulated in DLK1 KO cells, which might be expected based on the known associations between DLK1 and adrenocortical differentiation (see point 3 below). It follows that the correlation between expression levels of DLK1 and CYP17A1 should therefore be strongly positive in normal adrenal tissue (for example, in the GTEx adrenal dataset). However, by the same logic, one would similarly expect to find a strong positive correlation between DLK1 and ABCB1 expression levels if there is a previously undescribed, coregulation of these two genes also. We therefore highly recommend that the authors perform such an analysis, comparing the correlation between DLK1, CYP17A1 and ABCB1 in adrenal GTEx data (as they did with DLK1 and NOTCH1 in ACC), to establish whether the regulatory association proposed based on cell line experiments is biologically relevant by this criterion. If this turns out not to be the case, a stronger argument will need to be made to justify why decreased ABCB1 expression is not a technical artefact arising during DLK1 KO cell line preparation.

We have carefully considered the Reviewer's comments and feel that we may not have sufficient evidence to support the causal role of DLK1 in adrenocortical

differentiation (ADS). Thus, we have removed CYP17A1 and cortisol secretion data from Figure 5 (new Figure 6).

In regard to the relationship between DLK1, NOTCH1, and ABCB1, we would like to point out to the Reviewer that in addition to DLK1 KO data we also overexpressed the active, intracellular component of *NOTCH1* (i.e. *N1ICD*) in CU-ACC1 cells and observed a major downregulation of surface ABCB1 expression (Figure 6J). Moreover, unlike the DLK1 KO data, we did not perform clonal selection after *NOTCH1-ICD* overexpression, which we believe limits the probability of an artifact. Also, CU-ACC1 adherent cells with high NOTCH1 and low DLK1 expression showed lower expression of surface ABCB1 (Supplementary Figure 10G).

As per the Reviewer's suggestion, we analyzed both the ACC TCGA and GTEx adrenal dataset and found a negative correlation between *NOTCH1* and *ABCB1* expression (Figures 6K and 6L) thereby supporting our experimental data.

3) DLK1 in adrenocortical cell differentiation, stem/progenitor cells and ACC: The authors discuss the role of DLK1 as driver of adrenocortical cell differentiation and the relevance of this for the biology of ACC. However, they describe this out of the context of and without reference to a number of previously published results on the same topic. In these studies, the role of DLK1 in the homeostatic maintenance of the gland has already been investigated extensively. It has been proposed that the human adrenal cortex remodels with age to generate clusters of relatively undifferentiated cells expressing DLK1 (named DLK1-expressing cell clusters or DCCs), potentially representing a novel cell population in the human adrenal cortex.

Therefore, we recommend discussing the interpretation of any new data on this topic in the context of the following publications:

Guasti L, et al. Dlk1 up-regulates Gli1 expression in male rat adrenal capsule cells through the activation of $\beta 1$ integrin and ERK1/2. *Endocrinology*. 2013 doi: 10.1210/en.2013-1211.

Hadjidemetriou et al. DLK1/PREF1 marks a novel cell population in the human adrenal cortex. *J Steroid Biochem Mol Biol*. doi: 10.1016/j.jsbmb.2019.105422.

Pittaway et al. The role of delta-like non-canonical Notch ligand 1 (DLK1) in cancer. *Endocr Relat Cancer*. 2021 doi: 10.1530/ERC-21-0208.

Mariniello et al. Dlk1 is a novel adrenocortical stem/progenitor cell marker that predicts malignancy in adrenocortical carcinoma. *bioRxiv [Preprint]*. 2024 Aug 22:2024.08.22.609117. doi: 10.1101/2024.08.22.609117.

One of these (Hadjidemetriou et al) is cited on line 410 of the current manuscript but only in the context of levels of DLK1 expression in the adrenal cortex. We feel this does not do justice to any of the above listed studies that all put forward mechanistic roles for DLK1 in adrenal/ACC. In absence of appropriate referencing, this may give the reader a false impression any such claims are entirely novel to the current paper.

We appreciate these comments and agree with the Reviewer to discuss these prior studies. We have now added multiple sentences to the 2nd paragraph of the Discussion, referencing and discussing the 4 publications noted above: “In the adrenal cortex, DLK1 is widely expressed during development but postnatally is confined to undifferentiated cortical progenitor cells (Guasti et al. 2013). In cancer, DLK1 inhibits differentiation but has been shown to have both pro- and anti-proliferative effects (Pittaway et al. 2021) likely reflecting the complex relationship between DLK1 and NOTCH1 signaling across disease models (Huang et al. 2019; Zhong et al. 2017; Ranganathan et al. 2011). In ACC, while the role of DLK1 in tumorigenesis is still not well-defined, DLK1 expression is associated with worse recurrence-free survival (Marinello et al. 2025).”

Additionally, we have referenced the Marinello manuscript (which is now published) in regard DLK1 IHC staining in our Results section (as noted previously) and later in the 3rd paragraph of the Discussion section as follows: “Correspondingly, synaptophysin was observed to be highly upregulated in DLK1 high compared to DLK1 low ACC tumor regions using spatial transcriptomic analyses (Mariniello et al. 2025).”

As we mentioned earlier, since we are no longer included data related to DLK1 and adrenocortical differentiation (ADS), we have not discussed the above references in relation to ADS.

4) ACC patient tumour specimens and clinical data: We appreciate that the focus of the study is on advanced (metastatic) ACC. However, the authors should explain the

rationale behind their choice to not include any primary tumours, at least for a comparison with metastatic lesions.

While we primarily performed RNA-seq from metastatic tumors, we also performed RNA-seq from some archived primary tumor samples, which is now clearly shown in Supplementary Table 1 in Column H “sample type”. We have also clarified in the Methods section under “ACC patient tumor specimens” that our tissue collection cohort is primarily derived from metastatic surgical tissues: “ACC patient tumors in this study were collected under NIH Institutional Review Board protocols (NCT05237934, NCT01109394, and NCT03739827). Tumors were collected from surgical resection of metastatic sites at the NIH Clinical Center.”

Moreover, the authors only show the demographic and clinical data relative to the cohort used for short-term organoid experiments (n=13, Suppl Table 3). We recommend also displaying data for the entire cohort, i.e. including cases used for RNA-seq (n=46) and IHC (n=24). In fact, this information would be relevant for interpreting the biological relevance of the gene/protein expression data and response to treatment (e.g. evaluating the relationship between pre-treatment with mitotane or systemic drugs, and the steroid secretion profile, especially cortisol, or aggressiveness of the tumour etc.).

We have now added demographic and clinical data for the entire cohort: RNA-seq, IHC, and short-term organoid as suggested by the Reviewer, to Supplementary Table 1.

Additionally, the authors should also include among the presented data also the site of metastatic tumours tested in each experiment as well as the time passed from the primary surgery to the first disease recurrence and the surgery for metastatic disease.

In Supplementary Table 1 we have added “Time from primary surgery to sample collection (days)” in Column W and “biopsy or surgical site” in Column I.

Finally, regarding Suppl Table 3, there are few open issues to be improved/clarified, including: do the ki67% and the hormonal status data correspond to the primary tumour or metastasis? How many and which samples for each patient have been tested (the ID numbers do not always correlate with the colour code)? We strongly suggest revising this table to include such information.

In Supplementary Table 1 we have added “hormone producing tumors (yes/no)” in Column O and all variables including Ki67% (Column Q) are now listed for each sample site separately.

5) Bulk and single-cell RNA sequencing: In the methods section starting on line 461, the authors indicate that all bulk RNA-seq data generated in the current study involved

exclusively genetic material isolated from FFPE. Could they comment on the RNA-seq quality control for readers to assess the reliability of these results? To our knowledge, generating RNA-seq data from FFPE can be notoriously difficult and results must be carefully validated during quality control. Somewhat confusingly, immediately after this statement, the authors also say: “RNA-seq libraries were prepared using Illumina TruSeq RNA Access Library Prep Kit or Total RNA Library Prep Kit according to the manufacturer’s protocol (Illumina)”. Which one of these two kits was used and, if a combination of both, why was this so and what are the batch effects of using one kit compared to the other? Illumina’s own documentation highlights that not all their kits (e.g. TruSeq RNA Library Prep Kit v2) are compatible with FFPE extractions.

The Reviewer is correct that FFPE genetic material has previously posed challenges for RNA-sequencing. However, with advances in sequencing technology, FFPE is now widely used for RNA-seq. Specifically, Illumina has developed assays to capture the coding region of RNA directly as opposed to prior protocols in which RNA was poly-A tagged. This assay is called the TruSeq RNA Access Library Preparation Kit. Illumina also offers another library preparation kit called Total RNA Prep that can capture transcripts and non-coding RNA from FFPE-derived RNA. The TruSeq RNA Library Prep Kit v2 was not used for this study as that library is only warranted for polyA tagged mRNA.

There are also now a large number of clinical/translational studies which use FFPE samples for RNA-seq. See a few examples here (PubMed ID): 38366589, 35216676, 39577421. Caris Life Sciences has also sequenced thousands of tumors (both exome and RNA-seq) from FFPE genetic material using an FDA approved protocol: <https://www.carislifesciences.com/physicians/physician-tests/mi-cancer-seek/>

We used both the RNA Access Library and the Total RNA Library preps because our samples were sequenced under different clinical protocols. Since we observed a batch effect with these two methodologies (see below), we applied batch correction as stated in the Methods section: “We applied the "RemoveBatchEffect" function from the package Limma to remove the impact of the library preparation protocols (access or totalRNA).” Lastly, the quality of the FFPE RNA was assessed using DV200 value, which was performed by our Laboratory of Pathology. We have added a sentence in the Methods regarding this point: “RNA was first extracted by the NCI Laboratory of Pathology using CLIA certified procedures and all RNA had adequate DV₂₀₀ values suitable for sequencing.”

On several occasions (e.g. lines 323 and 470) the authors refer to the study Aber et al. manuscript in submission when, to our knowledge, no manuscript is available in either published or preprint form at the time of the current review. This is problematic, since the Aber et al study is used both to justify some major claims made in the current paper in addition to detailing the protocol for single-cell RNA-seq with metastatic ACC samples. Specifically, a claim is made by the authors that these single-cell data support the hypothesis that DLK1 expression is higher in populations of cells that are also enriched for the adrenal differentiation score (ADS), which is interpreted as a causal role for DLK1 in adrenocortical differentiation (this is emphasised on line 53-54 of the abstract, which indirectly references Aber et al, and is also related to our point 3 above). Beyond the lack of any publicly-available accompanying article by Aber et al to cross-refer to, we see two additional major problems with this: i) DLK1 itself is a member of the ADS gene set and so, by definition, there is covariation between its expression levels and the other genes within the set—simply removing DLK1 from the calculation of ADS presented in Fig 5K will not separate this (predefined) correlation from causation and such statistical comparisons to report p values appears unfounded (the ADS scores for DLK1 high and low groups are not necessarily independent);

As stated earlier, rather than focus on the relationship between DLK1 and adrenocortical differentiation (ADS), we feel our data are strongest to support the role of DLK1, through NOTCH1 signaling, in regulating *ABCB1* expression. Therefore, instead of showing the ADS score in relation to DLK1 expression from the single cell dataset, we now show that *ABCB1* expression is significantly higher in *NOTCH1* high compared to *NOTCH1* low single cells (Figure 6M).

M

ii) at the current time, we are aware of at least two other studies in published/preprint form that provide single-cell resolution data on ACC tumours but are not discussed/cited by the current study:

Tourigny et al Cellular landscape of adrenocortical carcinoma at single-nuclei resolution. Mol Cell Endocrinol. 2024 doi: 10.1016/j.mce.2024.112272

Popova et al Single nuclei sequencing reveals replication stress in adrenocortical carcinoma bioRxiv [Preprint]. 2024. doi: 10.1101/2024.09.30.615695

The first of these (Tourigny et al) investigates the role of the imprinted DLK1/MEG3 locus in ACC and the second (Popova et al) focusses specifically on advanced and metastatic ACC samples, as does Aber et al and the current study. At the very least, it would be highly appropriate to discuss the single cell results in the context of and with reference to these previous studies.

We agree with the Reviewer that discussing these two ACC single nuclei RNA-seq papers is important to include in our manuscript. We have included the following sentence in the 2nd paragraph of the Discussion:

“In two recent ACC single nuclei RNA-sequencing studies, DLK1 was highly expressed in clusters with abnormal DLK1 locus copy number states from primary surgical resected ACC tumors (Tourigny et al. 2024) and Hallmark Notch signaling pathway was suppressed across cell clusters from both primary and metastatic ACC tumors (Popova et al. 2024).”

6) Male/Female ratio: The authors utilised only female mice (n=7) and a large majority of female patients for the short-term organoid experiments (n=12 out of 13). Even if there is a slight preponderance of females among ACC patients, it is not observed at this level

(more than 90%). Moreover, it is known that pathogenic mechanisms involved in tumour development and progression might be different between males and females (i.e. through androgen-dependent induction of senescence, recruitment, and differentiation of highly phagocytic macrophages), see:

Wilmouth et al Sexually dimorphic activation of innate antitumor immunity prevents adrenocortical carcinoma development. *Sci Adv.* 2022 doi: 10.1126/sciadv.add0422.
Warde et al. Senescence-induced immune remodeling facilitates metastatic adrenal cancer in a sex-dimorphic manner. *Nat Aging.* 2023 Jul doi: 10.1038/s43587-023-00420-2.

We would like to ask the authors to consider whether relevant female preponderance could have affected the Notch/DLK1 expression data and/or the ADC *in vitro* and *in vivo* efficacy. The authors should justify their choice of focusing on females and discuss potential limitations. This is particularly relevant since DLK1 is an imprinted gene (an observation surprisingly not mentioned or discussed anywhere in the current study) and there are some existing theories that imprinting of maternal/paternal alleles may be directly related to selection for sex-specific function. See, for example:

Patten et al The evolution of genomic imprinting: theories, predictions and empirical tests. *Heredity* 2014 doi: 10.1038/hdy.2014.29

We agree with the Reviewer that our ACC cohort is likely more skewed towards female gender than other published ACC cohorts potentially because we function as a referral center for patients with metastatic disease interested in new clinical trials and/or surgical options.

Regarding the use of female vs. male mice: we have now clarified the sex of mice used in our experiments in the Methods section. We used female NSG mice for ADCT-701 *in vivo* experiments to correspond with the sex of our models (which are all derived from female patients). However, ACC PDXs were passaged in both male and female mice (depending on availability) and we have noted this now in the Methods section.

Nonetheless, we fully agree with the Reviewer that there is now ample evidence to suggest sex-specific functions driving ACC tumorigenesis. Still, as our experiments were not related to ACC tumorigenesis but rather therapeutic targeting, we do not believe that the sex of the host has a role in efficacy of this ADC. Moreover, the Marinello et al. manuscript did not find any correlation between DLK1 expression and sex or hormonal status. We have included the following sentence in the limitation section of the Discussion so the reader is aware we have considered this issue: "In light of recent studies implicating sexual dimorphism in ACC tumorigenesis (Wilmouth et al. and Warde et al.), we cannot rule out the possibility that sex may have a role in ADCT-701 efficacy as our experiments were primarily conducted using female patient-derived models;

however, DLK1 expression is not correlated with sex or hormonal status (Marinello et al. 2025)”.

We have also added to the Discussion a sentence in regard to DLK1 as a maternally imprinted, paternally expressed gene.

Minor comments:

- Introduction: add a reference for the sentence “recent FDA approvals across a diverse set of malignancies”

We have added the following reference for this sentence: Dumontet et al. Antibody-drug conjugates come of age in oncology. Nat Rev Drug Discov 2023.

- Adrenocortical Differentiation Score: We appreciate that ADS that is taken from the TCGA paper (Zheng et al 2016). However, a better description of the meaning and role of this score in this context would be useful

As noted above, we have removed mention of the Adrenocortical Differentiation Score in our manuscript.

- Discussion (page 14 line 364): a word after “driver of adrenocortical ...” is missing

We have modified this paragraph as we have now removed mention of the Adrenocortical Differentiation Score in our manuscript.

- One line 83, the authors say: “to our knowledge, there has been no systematic effort to assess whether Notch ligands beyond DLL3 may or may not be targetable cell surface proteins in cancer”. This sounds contradictory, since they cite reference 23 (preprint by Weiner et al now published as Hamilton et al Cancer Cell 2024 doi: 10.1016/j.ccell.2024.10.003), which has identified DLK1 as a surface target in neuroblastoma and has overlapping coauthors with the current study

We have modified this sentence as follows: “In this work, we sought to assess whether Notch ligands beyond DLL3 may or may not be targetable cell surface proteins in cancer.”

- We recommend updating the citation from above point to published version in Cancer Cell (author names have changed, which can add confusion)

We have updated this citation.

- At the end of the second results section, further cell death is proposed to occur via

“bystander killing” mediated by ADCT-701 treatment. Could the authors expand upon what this phenomenon is more generally, and speculate about possible mechanisms?

We have added more information regarding bystander killing effect in the Results section by including the following phrase: “...hydrophobic payloads such as our PBD can diffuse from target antigen-expressing cancer cells after direct ADC cytotoxicity into neighboring antigen-negative cancer cells...”

- On lines 179-181, it is not immediately intuitive how PDOs are classed as DLK1+ or DLK1- based on Figures 2G and H. Perhaps the authors could briefly clarify in that same statement the features of plots that allow this classification

We have modified the sentence to explain that a rightward shift in the flow cytometry histogram compared to unstained controls constitutes DLK1 positivity.

- In Figure 2I (and potentially 2J as well) the two plots appear to be different sizes even though the y axes should be the same. This should be corrected. It also makes direct comparison of the two cell lines difficult

We have corrected the sizes of Figure 2I/2J (which are now in a new Figure 3).

- Several references cited on page 14 seem to be mixed up and not corresponding to adequate papers (e.g. 30-36), which makes the reading difficult to fully understand. Please check

We have fixed the errors in the references.

- The word “differentiation” is missing after “adrenocortical” on line 364

As noted above, we have removed mention of the Adrenocortical Differentiation Score in our manuscript.

Reviewer #2 (Remarks to the Author): with expertise in antibody-drug conjugates, cancer therapy

Dear editor,

In this work by Yun Sun et al entitled “Identification of DLK1, a Notch ligand, as an immunotherapeutic target and regulator of tumor cell plasticity and chemoresistance in adrenocortical carcinoma” The authors describe the therapeutic activity of a novel humanized ADC against DLK1 in ACC and SCLC preclinical models and provide evidence that DLK1 notch ligand may represent a promising target for ADC therapy in these two particular indications, both of which represent hard to treat conditions. Moreover they undiscover the role of the transporter ABCB1 as major player in resistance mechanism to the ADC. Intriguingly, it’s proposed that the therapeutic target DLK1, is directly involved in the negative regulation of Notch signaling and positive regulation of ABCB1 expression which in turn increase chemoresistance. Therefore authors propose that this novel ADC-based therapeutic strategy may be restricted to DLK1 + tumors but with low/medium expression. Importantly, this novel ADC against DLK1 already entered in trial validation.

Overall, this is a solid work based on a consistent bulk of data. Indeed, generally experiments are well designed and conducted.

I would suggest publishing this manuscript once some aspects I raise here have been clarified:

We thank the Reviewer for the positive comments about our manuscript.

-A complete analytical characterization of this novel humanized ADC should be included in the manuscript

We appreciate this comment and have now included a more thorough characterization of ADCT-701 in the Results section as follows:

“ADCT-701 consists of HuBA-1-3D, a humanized anti-DLK1 monoclonal IgG1 kappa isotype antibody site-specifically conjugated Glycoconnect™ (van Geel et al. 2015) technology to the drug-linker PL1601, which contains HydraSpace™ (Verkade et al. 2018), a valine-alanine cleavable linker and the pyrrolobenzodiazepine (PBD) dimer SG3199 at a drug-to-antibody ratio of approximately 1.8 (Figure 2A). HuBA-1-3D was derived from the murine monoclonal antibody (mAb) BA-1-3D after it was humanized by grafting the complementarity determining region (CDR) of murine BA-1-3D into human IgG1 frameworks. Binding of HuBA-1-3D and ADCT-701 to human and cynomolgus monkey DLK1 showed similar affinities using surface plasma resonance analysis and no binding affinity was lost in ADCT-701 after conjugating HuBA-1-3D to the linker-payload (Hamilton et al. 2024).”

-The rationale behind the choice of dose and schedule relative to ADC treatment should be discussed.

We agree with the Reviewer. We used 1 mg/kg of ADCT-701 as a single dose based on a previously established *in vivo* dosing protocol for ADCT-701 (https://aacrjournals.org/cancerres/article/78/13_Supplement/744/630524/Abstract-744-ADCT-701-a-novel). We have stated the dose and schedule of our ADC treatments in the Results section and cited the above reference in both the Results and Methods sections.

-Figure 2C/4E: ADCT701 *in vitro* cell killing activity is shown. IC50 calculation is missing.

We appreciate this correction from the Reviewer. We have added the IC50 calculation in Figure 2C/4E (which are now Figure 2D/5E).

-Figure 3A-C. Authors clearly demonstrated that there's an upregulation of ABCB1 transporter gene in ACC patients. In parallel, this upregulation has been confirmed in insensitive but not sensitive cell lines thus suggesting that ABCB1 upregulation may be involved in SG3199 resistance. However, authors should explain why ACC48 cells has been selected as principal model for all the further experiments and not ACC40 cell line model given the fact that the former one resulted as the most insensitive to the ADC. Moreover, ACC40 cells do express very high amount of DLK1 (Figure 2H). Is, as expected, ABCB1 upregulated in these cells?

We agree with the Reviewer that additional models would strengthen our findings. Thus, we have added new experimental data with NCI-ACC40 cells, namely surface ABCB1 expression, and ADCT-701/PBD with or without ABCB1 inhibitors cytotoxicity in Figure 4C and Figure 4D/Supplementary Figure 7B, respectively. We found that ABCB1 inhibitors dramatically increase sensitivity to ADCT-701 and PBD in NCI-ACC40 in concordance with data from NCI-ACC48.

-Figure 3J: statistic is missing. Indeed, re-sensitization to PBD by ABCB1 inhibitors seems not to be so relevant here. Authors are invited to comment on this.

We thank the Reviewer for their comment. We previously included re-sensitization to ADCT-701 by ABCB1 inhibitors in Supplementary Fig. 7 and Supplementary Fig. 8. We have now moved that data to the new Figure 4 (prior Figure 3) and the re-sensitization to PBD by ABCB1 inhibitors is now in Supplementary Fig. 7 and Supplementary Fig. 8. Together, these data are important because they functionally demonstrate the role of the ABCB1 drug efflux transporter in mediating both intrinsic and acquired resistance to ADCT-701.

-Figure 5. Immunoblot and/or mRNA data showing DLK1 knock-out in the selected clones are missing.

We would like to point out to the Reviewer that both immunoblot and flow cytometry data showing lack of DLK1 expression in DLK1 KO CU-ACC1 clones were shown in Supplementary Figure 2B, D. However, given the importance of showing to the reader that we successfully performed DLK1 KO, we have now moved the immunoblot data to Figure 6A.

Figure 6

To validate DLK1KO data authors used DLK+ ACC48 and DLK- ACC49 cell models. Again, ACC40 cell model would be useful here, as these cells express very high amount of DLK1. Including a second DLK1 high expressing model would reinforce author's findings.

We have now added immunoblot data for NCI-ACC40 together with ACC48 and ACC49 models (Figure 6G). We observed higher expression of N1ICD and lower expression of SYP in the DLK1 negative NCI-ACC49 PDO compared to the DLK1 positive NCI-ACC40 and NCI-ACC48 PDOs.

In view of the potential relevance of the role of DLK1 in inducing ADC and chemoresistance, it would be of great interest to evaluate this phenomenon in other than rare ACC tumors. In particular, as example it is known that in ovarian cancer overexpression of DLK1 is associated with a worse prognosis. It would be important to verify whether regulation of ABCB1 by DLK1 is also found in this type of tumor. This information will define whether the mechanisms described here are limited to ACC tumors or shared by other DLK+ malignancies.

Per the editor, we do not need to address this comment. However, we have noted in the Discussion that Notch signaling is highly context dependent in various tumor types and thus our findings should not be extrapolated beyond ACC (and likely other neuroendocrine tumors).

Reviewer #3 (Remarks to the Author): with expertise in adrenocortical tumors

In this paper, the authors identify DLK1 as a potential immunotherapeutic target in ACC and describe some insights into the role of this gene in tumor biology and chemoresistance. Although similar data, using the same ADC and similar methodology, have recently been published in neuroblastoma (Cancer Cell, <https://doi.org/10.1016/j.ccell.2024.10.003>), which I suggest should be added to the manuscript discussion and referenced, the manuscript presents for the first time consistent data in ACC cell lines. The manuscript is clearly written, uses appropriate methodology to support the conclusions, and provides interesting information to the readers of Nature Communication.

We very much thank the Reviewer for these positive comments regarding our manuscript. We did reference the DLK1 neuroblastoma paper in the Results and in the Discussion sections. However, when we submitted our manuscript, this paper was still only available on the bioRxiv. We have now switched all references to the published paper (Hamilton et al. Cancer Cell).

I have a few points to address:

1) The DLK1 gene is paternally expressed and related to the Dlk1-Dio3 imprinted domain and regulated by a complex epigenetic process controlled by an imprinted control region (IG-DMR) that is hypomethylated on the maternal allele and hypermethylated on the paternal allele (doi.org/10.1093/nar/gkac344). I think it is important to describe better this genetic particularity and how it may interfere with tumor biology and potential treatment.

We fully agree with the Reviewer. We have now included a sentence regarding DLK1 imprinting in the 2nd paragraph of the Discussion: “DLK1 is maternally imprinted, paternally expressed gene within the *DLK1-DIO3* gene cluster (Kojima et al. 2022) that encodes a transmembrane glycoprotein with a structure similar to canonical Notch ligands.”

We have also included a sentence regarding this in relation to treatment in the limitations section of the Discussion: “Apart from ACC, we found to DLK1 is more heterogeneously expressed in neuroendocrine cancers such as SCLC suggesting further insights into mechanisms of DLK1 expression, particularly potential epigenetic mechanisms that may control DLK1 imprinting, may be important for the development of potential future DLK1-targeted combination therapies.”

2) Although DLK1 is expressed in many human tissues during embryonic development, in adults its expression is mainly restricted to (neuro)endocrine tissues and other

immature stem/progenitor cells, being important in several physiological mechanisms. In my opinion, the potential deleterious effects of an ADC against this gene could be discussed.

We agree that including information on potential deleterious effects on immature stem/progenitor cells by targeting DLK1 is important. In the Discussion section on potential DLK1 toxicity we have included the following sentence: “Targeting DLK1 could have also have deleterious effects on normal tissue or organ regeneration as DLK1 is expressed on and regulates many immature stem/progenitor cells such as hepatoblasts (Grassi et al. 2022).”

3) The evidence for the nature of the effect of DLK1 on Notch signaling is conflicting, with some results opposing those described in this manuscript. In high-grade serous carcinoma of the ovary (<https://doi.org/10.1038/s41388-018-0658-5>) and in NSCLC cell models (10.3892/ol.2017.6019), unlike what was found by the authors, a positive relationship between DLK1 and NOTCH1 was described. I think it would be interesting to describe in the SCLC model used the relationship between DLK1 and NOTCH1 and further discuss the confliction association between these genes.

We agree with this comment and have now add a sentence including the references cited by the Reviewer: “In cancer, DLK1 inhibits differentiation but has been shown to have both pro- and anti-proliferative effects (Pittaway et al. 2021) likely reflecting the complex relationship between DLK1 and NOTCH1 signaling across disease models (Huang et al. 2019; Zhong et al. 2017; Ranganathan et al. 2011).”

Given that the focus of the manuscript is ACC, we did not perform DLK1 KO in a SCLC model; therefore, we cannot definitively state whether DLK1 positively or negative regulates NOTCH1 in SCLC. However, we have now interrogated the relationship between NOTCH1 and DLK1 in SCLC using RNA-seq data from SCLC primary tumors (George et al. Nature 2025). In this dataset, we found a strong, negative correlation between DLK1 and NOTCH1 consistent with our data in ACC (Supplementary Figure 9D).

4) The hPheo1 cell line described in lines 139-141 and supplementary figure 2 is not described in the methods

We thank the Reviewer for this correction. Within the “Cell lines”, “Flow cytometric analysis” and “In vitro cell line cytotoxicity assays” section of the Methods we have added in language describing our work with hPheo1.

5) Lines 170-171. The authors described that “Overall, these results indicate that ADCT-701 not only can target DLK1+ cells but can also indirectly induce cytotoxicity in DLK1- cells”. This potential indirect mechanism can be further discussed.

We have now clarified in this sentence that “indirectly inducing cytotoxicity in DLK1- cells” is through bystander killing.

The same goes for lines 180-181: “However, among ADCT-701 non-responders, 50% (n=3/6) were still DLK1+ (Fig. 2H) suggesting that ADCT-701 sensitivity is influenced by factors other than DLK1 expression”. Which factors?

We have now modified this sentence to make our point clearer. Instead of “ADCT-701 sensitivity is influenced by factors other than DLK1 expression” we state, “the lack of target antigen expression may not be a mechanism of resistance to ADCT-701.” In the next section of the Results, we address mechanisms of resistance to ADCT-701 in our DLK1+ ACC pre-clinical models.

6) It is not clear from the text whether all in vitro experiments were performed in at least three independent experiments, in triplicate. In some of them, the SD bars in the figures are not visible: Fig. 2 C and E (for NCI-ACC-47); Fig. 2 F (for NCI-ACC-48); Fig. 3 D and F; Fig. 4 D and E (for H524); Fig. 5 D; Supplementary Fig. 2 A and D; Supplementary Fig. 3 E, Supplementary Fig. 7 E, Supplementary Fig. 10 I and J (for VP16 and doxo) and K. Please clarify.

We appreciate this important comment from the Reviewer. We have now added information regarding the biological and technical replicates in all Figure legends. For *in vitro* cell line experiments, data shown are mean \pm SEM of three or four independent experiments. For *in vitro* organoid experiments, data are mean \pm SEM of three technical replicates (n=1 or 2 biological replicates). We were not able to perform multiple independent experiments in some of the short-term ACC organoid models.

7) There are some errors and inconsistencies in the figures. Figure 1E is not described in the figure legend. Figure 3 G, please standardize the formatting and drug description. Supplementary Fig. 10, the letters did not match in the figure, e.g., legend “(D) Photomicrographs of CU-ACC1 parental and DLK KO clones 9 and 10 cells”, in Figure D is an anti-ABCB1 histogram. Figure K is not described in the legend. Please review.

We appreciate the Reviewer for pointing out these errors. We have added Figure 1E description in the legend. We have standardized the formatting and drug description in Figure 3G (now Figure 4G) and fixed the legend for Supplementary Fig. 10.

Reviewer #4 (Remarks to the Author):

REVIEWERS' COMMENTS

We would like to thank all the Reviewers for their constructive comments on our manuscript.

Reviewer #1 (Remarks to the Author):

The Authors have extensively revised the manuscript according to reviewer's requests and satisfactorily replied to the large majority of my comments. I appreciate that the current version is significantly improved with clearer conclusions. I only have a minor request, i.e. to add to add a comment regarding the DLK1 antibody as short sentence within the limitations of the study.

We have now added the following sentence to the limitations section regarding the DLK1 antibody: "Of note, given the challenge of assessing membrane-specific DLK1 expression with current antibodies, alternative detection methods may be required for future DLK1 targeting clinical trials."

Reviewer #2 (Remarks to the Author):

Dear Editor,

In my opinion this manuscript can be published in the current form. Authors have successfully addressed all the criticisms raised.

Reviewer #3 (Remarks to the Author):

I consider all questions and suggestions made by this reviewer to have been adequately answered in the revised form, with no new comments or questions.

Reviewer #4 (Remarks to the Author):
